# EE-Net: Exploitation-Exploration Neural Networks in Contextual Bandits

**Yikun Ban, Yuchen Yan, Arindam Banerjee, Jingrui He**
University of Illinois at Urbana-Champaign
`{yikunb2, yucheny5, arindamb, jingrui}@illinois.edu`

## Abstract

In this paper, we propose a novel neural exploration strategy in contextual bandits, EE-Net, distinct from the standard UCB-based and TS-based approaches. Contextual multi-armed bandits have been studied for decades with various applications. To solve the exploitation-exploration tradeoff in bandits, there are three main techniques: epsilon-greedy, Thompson Sampling (TS), and Upper Confidence Bound (UCB). In recent literature, linear contextual bandits have adopted ridge regression to estimate the reward function and combine it with TS or UCB strategies for exploration. However, this line of works explicitly assumes the reward is based on a linear function of arm vectors, which may not be true in real-world datasets. To overcome this challenge, a series of neural bandit algorithms have been proposed, where a neural network is used to learn the underlying reward function and TS or UCB are adapted for exploration. Instead of calculating a large-deviation based statistical bound for exploration like previous methods, we propose "EE-Net", a novel neural-based exploration strategy. In addition to using a neural network (Exploitation network) to learn the reward function, EE-Net uses another neural network (Exploration network) to adaptively learn potential gains compared to the currently estimated reward for exploration. Then, a decision-maker is constructed to combine the outputs from the Exploitation and Exploration networks. We prove that EE-Net can achieve $\mathcal{O}(\sqrt{T \log T})$ regret and show that EE-Net outperforms existing linear and neural contextual bandit baselines on real-world datasets.

## 1 Introduction

The stochastic contextual multi-armed bandit (MAB) (Lattimore and Szepesvári, 2020) has been studied for decades in machine learning community to solve sequential decision making, with applications in online advertising (Li et al., 2010), personal recommendation (Wu et al., 2016; Ban and He, 2021b), etc. In the standard contextual bandit setting, a set of $n$ arms are presented to a learner in each round, where each arm is represented by a context vector. Then by certain strategy, the learner selects and plays one arm, receiving a reward. The goal of this problem is to maximize the cumulative rewards of $T$ rounds.

MAB algorithms have principled approaches to address the trade-off between Exploitation and Exploration (EE), as the collected data from past rounds should be exploited to get good rewards but also under-explored arms need to be explored with the hope of getting even better rewards. The most widely-used approaches for EE trade-off can be classified into three main techniques: Epsilon-greedy (Langford and Zhang, 2008), Thompson Sampling (TS) (Thompson, 1933), and Upper Confidence Bound (UCB) (Auer, 2002; Ban and He, 2020).

Linear bandits (Li et al., 2010; Dani et al., 2008; Abbasi-Yadkori et al., 2011), where the reward is assumed to be a linear function with respect to arm vectors, have been well studied and succeeded both empirically and theoretically. Given an arm, ridge regression is usually adapted to estimate its reward based on collected data from past rounds. UCB-based algorithms (Li et al., 2010; Chu et al., 2011; Wu et al., 2016; Ban and He, 2021b) calculate an upper bound for the confidence ellipsoid of estimated reward and determine the arm according to the sum of estimated reward and UCB. TS-based algorithms (Agrawal and Goyal, 2013; Abeille and Lazaric, 2017) formulate each arm as a posterior distribution where mean is the estimated reward and choose the one with the maximal

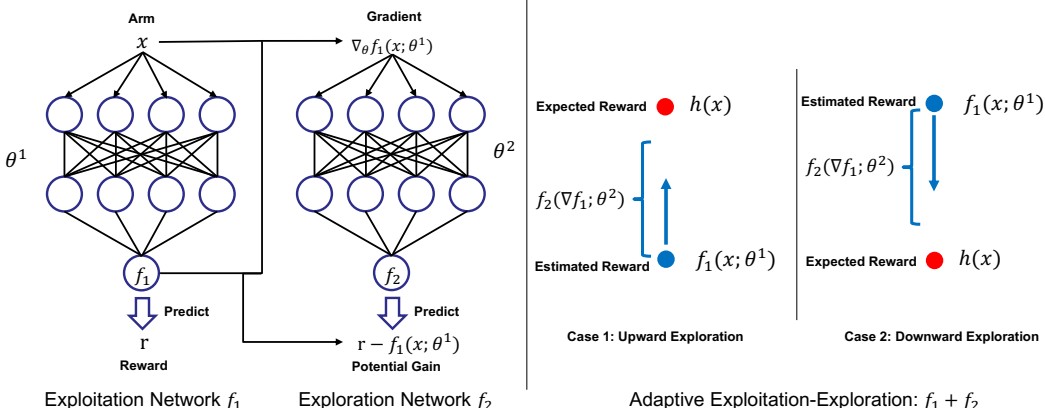

Figure 1: Left figure: Structure of EE-Net. In the right figure, Case 1: "Upward" exploration should be made when the learner underestimates the reward; Case 2: "Downward" exploration should be chosen when the learner overestimates the reward. EE-Net has the ability to adaptively make exploration according to different cases. In contrast, UCB-based strategy will always make upward exploration, and TS-based strategy will randomly choose upward or downward exploration.

sampled reward. However, the linear assumption regarding the reward may not be true in real-world applications (Valko et al., 2013).

To learn non-linear reward functions, recent works have utilized deep neural networks to learn the underlying reward functions, thanks to its powerful representation ability. Considering the past selected arms and received rewards as training samples, a neural network $f_1$ is built for exploitation. Zhou et al. (2020) computes a gradient-based upper confidence bound with respect to $f_1$ and uses UCB strategy to select arms. Zhang et al. (2021) formulates each arm as a normal distribution where the mean is $f_1$ and deviation is calculated based on gradient of $f_1$, and then uses the TS strategy to choose arms. Both Zhou et al. (2020) and Zhang et al. (2021) achieve the near-optimal regret bound of $O(\sqrt{T}\log T)$.

In this paper, we propose a novel neural exploration strategy, named "EE-Net". Similar to other neural bandits, EE-Net has another exploitation network $f_1$ to estimate rewards for each arm. The crucial difference from existing works is that EE-Net has an exploration network $f_2$ to predict the potential gain for each arm compared to current reward estimate. The input to the exploration network is the gradient of $f_1$ and the ground-truth is residual difference between the true received reward and the estimated reward from $f_1$. The strategy is inspired by recent advances in the neural UCB strategies (Zhou et al., 2020; Ban et al., 2021). Finally, a decision-maker $f_3$ is constructed to select arms. $f_3$ has two modes: linear or nonlinear. In linear mode, $f_3$ is a linear combination of $f_1$ and $f_2$, inspired by the UCB strategy. In the nonlinear mode, $f_3$ is formulated as a neural network with input $(f_1, f_2)$ and the goal is to learn the probability of being an optimal arm for each arm. Figure 1 depicts the workflow of EE-Net and its advantages for exploration compared to UCB or TS-based methods (see more details in Appendix D). To sum up, the contributions of this paper can be summarized as follows:

1. We propose a novel neural exploration strategy, EE-Net, where another neural network is assigned to learn the potential gain compared to the current reward estimate.

2. Under standard assumptions of over-parameterized neural networks, we prove that EE-Net can achieve the regret upper bound of $\mathcal{O}(\sqrt{T\log T})$, which improves a multiplicative factor of $\sqrt{\log T}$ and is independent of either input or effective dimension, compared to existing state-of-the-art neural bandit algorithms.

3. We conduct extensive experiments on four real-world datasets, showing that EE-Net outperforms baselines including linear and neural versions of $\epsilon$-greedy, TS, and UCB.

Next, we discuss the problem definition in Sec.3, elaborate on the proposed EE-Net in Sec.4, and present our theoretical analysis in Sec.5. In the end, we provide the empirical evaluation (Sec.6) and conclusion.

## 2 RELATED WORK

**Constrained Contextual bandits**. The common constrain placed on the reward function is the linear assumption, usually calculated by ridge regression (Li et al., 2010; Abbasi-Yadkori et al., 2011; Valko et al., 2013; Dani et al., 2008). The linear UCB-based bandit algorithms (Abbasi-Yadkori et al., 2011; Li et al., 2016) and the linear Thompson Sampling (Agrawal and Goyal, 2013; Abeille and Lazaric, 2017) can achieve successful performance and the near-optimal regret bound of $\tilde{\mathcal{O}}(\sqrt{T})$. To break the linear assumption, Filippi et al. (2010) generalizes the reward function to a composition of linear and non-linear functions and then adopt a UCB-based algorithm to deal with it; Bubeck et al. (2011) imposes the Lipschitz property on reward metric space and constructs a hierarchical optimistic optimization to make selections; Valko et al. (2013) embeds the reward function into Reproducing Kernel Hilbert Space and proposes the kernelized TS/UCB bandit algorithms.

**Neural Bandits**. To learn non-linear reward functions, deep neural networks have been adapted to bandits with various variants. Riquelme et al. (2018); Lu and Van Roy (2017) build L-layer DNN to learn the arm embeddings and apply Thompson Sampling on the last layer for exploration. Zhou et al. (2020) first introduces a provable neural-based contextual bandit algorithm with a UCB exploration strategy and then Zhang et al. (2021) extends the neural network to Thompson sampling framework. Their regret analysis is built on recent advances on the convergence theory in over-parameterized neural networks(Du et al., 2019; Allen-Zhu et al., 2019) and utilizes Neural Tangent Kernel (Jacot et al., 2018; Arora et al., 2019) to construct connections with linear contextual bandits (Abbasi-Yadkori et al., 2011). Ban and He (2021a) further adopts convolutional neural networks with UCB exploration aiming for visual-aware applications. Xu et al. (2020) performs UCB-based exploration on the last layer of neural networks to reduce the computational cost brought by gradient-based UCB. Different from the above existing works, EE-Net keeps the powerful representation ability of neural networks to learn reward function and first assigns another neural network to determine exploration.

## 3 PROBLEM DEFINITION

We consider the standard contextual multi-armed bandit with the known number of rounds $T$ (Zhou et al., 2020; Zhang et al., 2021). In each round $t \in [T]$, where the sequence $[T] = [1, 2, \ldots, T]$, the learner is presented with $n$ arms, $\mathbf{X}_t = \{\mathbf{x}_{t,1}, \ldots, \mathbf{x}_{t,n}\}$, in which each arm is represented by a feature vector $\mathbf{x}_{t,i} \in \mathbb{R}^d$ for each $i \in [n]$. After playing one arm $\mathbf{x}_{t,i}$, its reward $r_{t,i}$ is assumed to be generated by the function:

$$r_{t,i} = h(\mathbf{x}_{t,i}) + \eta_{t,i}, \tag{3.1}$$

where the *unknown* reward function $h(\mathbf{x}_{t,i})$ can be either linear or non-linear and the noise $\eta_{t,i}$ is drawn from certain distribution with expectation $\mathbb{E}[\eta_{t,i}] = 0$. Following many existing works (Zhou et al., 2020; Ban et al., 2021; Zhang et al., 2021), we consider bounded rewards, $r_{t,i} \in [a, b]$. For the brevity, we denote the selected arm in round $t$ by $\mathbf{x}_t$ and the reward received in $t$ by $r_t$. The pseudo *regret* of $T$ rounds is defined as:

$$\mathbf{R}_T = \mathbb{E}\left[\sum_{t=1}^{T}(r_t^* - r_t)\right], \tag{3.2}$$

where $\mathbb{E}[r_t^* \mid \mathbf{X}_t] = \max_{i \in [n]} h(\mathbf{x}_{t,i})$ is the maximal expected reward in the round $t$. The goal of this problem is to minimize $\mathbf{R}_T$ by certain selection strategy.

**Notation.** We denote by $\{\mathbf{x}_i\}_{i=1}^{t}$ the sequence $(\mathbf{x}_1, \ldots, \mathbf{x}_t)$. We use $\|v\|_2$ to denote the Euclidean norm for a vector $v$, and $\|\mathbf{W}\|_2$ and $\|\mathbf{W}\|_F$ to denote the spectral and Frobenius norm for a matrix $\mathbf{W}$. We use $\langle \cdot, \cdot \rangle$ to denote the standard inner product between two vectors or two matrices. We may use $\nabla_{\boldsymbol{\theta}_t^1} f_1(\mathbf{x}_{t,i})$ or $\nabla_{\boldsymbol{\theta}_t^1} f_1$ to represent the gradient $\nabla_{\boldsymbol{\theta}_t^1} f_1(\mathbf{x}_{t,i}; \boldsymbol{\theta}_t^1)$ for brevity. We use $\{\mathbf{x}_\tau, r_\tau\}_{\tau=1}^{t}$ to represent the collected data up to round $t$.

## 4 PROPOSED METHOD: EE-NET

EE-Net is composed of three components. The first component is the *exploitation network*, $f_1(\cdot; \boldsymbol{\theta}^1)$, which focuses on learning the unknown reward function $h$ based on the data collected in past rounds.

Table 1: Structure of EE-Net (Round $t$).

| Input | Network | Label |
|---|---|---|
| $\{\mathbf{x}_\tau\}_{\tau=1}^t$ | $f_1(\cdot; \boldsymbol{\theta}^1)$ (Exploitation) | $\{r_\tau\}_{\tau=1}^t$ |
| $\{\nabla_{\boldsymbol{\theta}_{\tau-1}^1} f_1(\mathbf{x}_\tau; \boldsymbol{\theta}_{\tau-1}^1)\}_{\tau=1}^t$ | $f_2(\cdot; \boldsymbol{\theta}^2)$ (Exploration) | $\left\{\left(r_\tau - f_1(\mathbf{x}_\tau; \boldsymbol{\theta}_{\tau-1}^1)\right)\right\}_{\tau=1}^t$ |
| $\{(f_1(\mathbf{x}_\tau; \boldsymbol{\theta}_{\tau-1}^1), f_2(\nabla_{\boldsymbol{\theta}_{\tau-1}^1} f_1; \boldsymbol{\theta}_{\tau-1}^2))\}_{\tau=1}^t$ | $f_3(\cdot; \boldsymbol{\theta}^3)$ (Decision-maker with non-linear function) | $\{p_\tau\}_{\tau=1}^t$ |

The second component is the *exploration network*, $f_2(\cdot; \boldsymbol{\theta}^2)$, which focuses on characterizing the level of exploration needed for each arm in the present round. The third component is the *decision-maker*, $f_3$, which focuses on suitably combining the outputs of the exploitation and exploration networks leading to the arm selection.

**1) Exploitation Net.** The exploitation net $f_1$ is a neural network which learns the mapping from arms to rewards. In round $t$, denote the network by $f_1(\cdot; \boldsymbol{\theta}_{t-1}^1)$, where the superscript of $\boldsymbol{\theta}_{t-1}^1$ is the index of network and the subscript represents the round where the parameters of $f_1$ finished the last update. Given an arm $\mathbf{x}_{t,i}, i \in [n]$, $f_1(\mathbf{x}_{t,i}; \boldsymbol{\theta}_{t-1}^1)$ is considered the "exploitation score" for $\mathbf{x}_{t,i}$. By some criterion, after playing arm $\mathbf{x}_t$, we receive a reward $r_t$. Therefore, we can conduct gradient descent to update $\boldsymbol{\theta}^1$ based on the collected training samples $\{\mathbf{x}_\tau, r_\tau\}_{\tau=1}^t$ and denote the updated parameters by $\boldsymbol{\theta}_t^1$.

**2) Exploration Net.** Our exploration strategy is inspired by existing UCB-based neural bandits (Zhou et al., 2020; Ban et al., 2021). Based on the Lemma 5.2 in (Ban et al., 2021), given an arm $\mathbf{x}_{t,i}$, with probability at least $1 - \delta$, we have the following UCB form:

$$|h(\mathbf{x}_{t,i}) - f_1(\mathbf{x}_{t,i}; \boldsymbol{\theta}_{t-1}^1)| \leq \Psi(\nabla_{\boldsymbol{\theta}_{t-1}^1} f_1(\mathbf{x}_{t,i}; \boldsymbol{\theta}_{t-1}^1)), \tag{4.1}$$

where $h$ is defined in Eq. (3.1) and $\Psi$ is an upper confidence bound represented by a function with respect to the gradient $\nabla_{\boldsymbol{\theta}_{t-1}^1} f_1$ (see more details and discussions in Appendix D). Then we have the following definition.

**Definition 4.1.** *In round $t$, given an arm $\mathbf{x}_{t,i}$, we define $h(\mathbf{x}_{t,i}) - f_1(\mathbf{x}_{t,i}; \boldsymbol{\theta}_{t-1}^1)$ as the "expected potential gain" for $\mathbf{x}_{t,i}$ and $r_{t,i} - f_1(\mathbf{x}_{t,i}; \boldsymbol{\theta}_{t-1}^1)$ as the "potential gain" for $\mathbf{x}_{t,i}$.*

Let $y_{t,i} = r_{t,i} - f_1(\mathbf{x}_{t,i}; \boldsymbol{\theta}_{t-1}^1)$. When $y_{t,i} > 0$, the arm $\mathbf{x}_{t,i}$ has positive potential gain compared to the estimated reward $f_1(\mathbf{x}_{t,i}; \boldsymbol{\theta}_{t-1}^1)$. A large positive $y_{t,i}$ makes the arm more suitable for exploration, whereas a small (or negative) $y_{t,i}$ makes the arm unsuitable for exploration. Recall that traditional approaches such as UCB intend to estimate such potential gain $y_{t,i}$ using standard tools, e.g., Markov inequality, Hoeffding bounds, etc., from large deviation bounds.

Instead of calculating a large-deviation based statistical bound for $y_{t,i}$, we use a neural network $f_2(\cdot; \boldsymbol{\theta}^2)$ to represent $\Psi$, where the input is $\nabla_{\boldsymbol{\theta}_{t-1}^1} f_1(\mathbf{x}_{t,i})$ and the ground truth is $r_{t,i} - f_1(\mathbf{x}_{t,i}; \boldsymbol{\theta}_{t-1}^1)$. Adopting gradient $\nabla_{\boldsymbol{\theta}_{t-1}^1} f_1(\mathbf{x}_{t,i})$ as the input also is due to the fact that it incorporates two aspects of information: the feature of the arm and the discriminative information of $f_1$.

Moreover, in the upper bound of NeuralUCB or the variance of NeuralTS, there is a recursive term $\mathbf{A}_{t-1} = \mathbf{I} + \sum_{\tau=1}^{t-1} \nabla_{\boldsymbol{\theta}_{\tau-1}^1} f_1(\mathbf{x}_\tau) \nabla_{\boldsymbol{\theta}_{\tau-1}^1} f_1(\mathbf{x}_\tau)^\top$ which is a function of past gradients up to $(t-1)$ and incorporates relevant historical information. On the contrary, in EE-Net, the recursive term which depends on past gradients is $\boldsymbol{\theta}_{t-1}^2$ in the exploration network $f_2$ because we have conducted gradient descent for $\boldsymbol{\theta}_{t-1}^2$ based on $\{\nabla_{\boldsymbol{\theta}_{\tau-1}^1} f_1(\mathbf{x}_\tau)\}_{\tau=1}^{t-1}$. Therefore, this form $\boldsymbol{\theta}_{t-1}^2$ is similar to $\mathbf{A}_{t-1}$ in neuralUCB/TS, but EE-net does not (need to) make a specific assumption about the functional form of past gradients, and is also more memory-efficient.

To sum up, in round $t$, we consider $f_2(\nabla_{\boldsymbol{\theta}_{t-1}^1} f_1(\mathbf{x}_{t,i}); \boldsymbol{\theta}_{t-1}^2)$ as the "exploration score" of $\mathbf{x}_{t,i}$, because it indicates the potential gain of $\mathbf{x}_{t,i}$ compared to our current exploitation score $f_1(\mathbf{x}_{t,i}; \boldsymbol{\theta}_{t-1}^1)$. Therefore, after receiving the reward $r_t$, we can use gradient descent to update $\boldsymbol{\theta}^2$ based on collected training samples $\{\nabla_{\boldsymbol{\theta}_{\tau-1}^1} f_1(\mathbf{x}_\tau), r_\tau - f_1(\mathbf{x}_\tau; \boldsymbol{\theta}_{\tau-1}^1)\}_{\tau=1}^t$. We also provide other two heuristic forms for $f_2$'s ground-truth label: $|r_{t,i} - f_1(\mathbf{x}_{t,i}; \boldsymbol{\theta}_{t-1}^1)|$ and $\text{ReLU}(r_{t,i} - f_1(\mathbf{x}_{t,i}; \boldsymbol{\theta}_{t-1}^1))$. We compare them in an ablation study in Appendix B.

---

**Algorithm 1** EE-Net

---

**Input:** $f_1, f_2, f_3$, $T$ (number of rounds), $\eta_1$ (learning rate for $f_1$), $\eta_2$ (learning rate for $f_2$), $\eta_3$ (learning rate for $f_3$), $K_1$ (number of iterations for $f_1$), $K_2$ (number of iterations for $f_2$) , $K_3$ (number of iterations for $f_3$), $\phi$ (normalization operator)

1: Initialize $\boldsymbol{\theta}_0^1, \boldsymbol{\theta}_0^2, \boldsymbol{\theta}_0^3; \widehat{\boldsymbol{\theta}}_0^1 = \boldsymbol{\theta}_0^1, \widehat{\boldsymbol{\theta}}_0^2 = \boldsymbol{\theta}_0^2, \widehat{\boldsymbol{\theta}}_0^3 = \boldsymbol{\theta}_0^3$
2: **for** $t = 1, 2, \dots, T$ **do**
3:     Observe $n$ arms $\{\mathbf{x}_{t,1}, \dots, \mathbf{x}_{t,n}\}$
4:     **for** each $i \in [n]$ **do**
5:         Compute $f_1(\mathbf{x}_{t,i}; \boldsymbol{\theta}_{t-1}^1), f_2(\phi(\nabla_{\boldsymbol{\theta}_{t-1}^1} f_1(\mathbf{x}_{t,i})); \boldsymbol{\theta}_{t-1}^2), f_3((f_1, f_2); \boldsymbol{\theta}_{t-1}^3)$
6:     **end for**
7:     $\mathbf{x}_t = \arg\max_{\mathbf{x}_{t,i}, i \in [n]} f_3\left(f_1(\mathbf{x}_{t,i}; \boldsymbol{\theta}_{t-1}^1), f_2(\phi(\nabla_{\boldsymbol{\theta}_{t-1}^1} f_1(\mathbf{x}_{t,i})); \boldsymbol{\theta}_{t-1}^2); \boldsymbol{\theta}_{t-1}^3\right)$
8:     Play $\mathbf{x}_t$ and observe reward $r_t$
9:     $\boldsymbol{\theta}_t^1, \boldsymbol{\theta}_t^2, \boldsymbol{\theta}_t^3 = \text{GRADIENTDESCENT}(\boldsymbol{\theta}_0, \{\mathbf{x}_\tau\}_{\tau=1}^t, \{r_\tau\}_{\tau=1}^t)$
10: **end for**
11:
12: **procedure** GRADIENTDESCENT($\boldsymbol{\theta}_0, \{\mathbf{x}_\tau\}_{\tau=1}^t, \{r_\tau\}_{\tau=1}^t$)
13:     $\mathcal{L}_1 = \frac{1}{2} \sum_{\tau=1}^t \left(f_1(\mathbf{x}_\tau; \boldsymbol{\theta}^1) - r_\tau\right)^2$
14:     $\boldsymbol{\theta}^{1,(0)} = \boldsymbol{\theta}_0^1$
15:     **for** $k \in \{1, \dots, K_1\}$ **do**
16:         $\boldsymbol{\theta}^{1,(k)} = \boldsymbol{\theta}^{1,(k-1)} - \eta_1 \nabla_{\boldsymbol{\theta}^{1,(k-1)}} \mathcal{L}_1$
17:     **end for**
18:     $\widehat{\boldsymbol{\theta}}_t^1 = \boldsymbol{\theta}^{1,(K_1)}$
19:     $\mathcal{L}_2 = \frac{1}{2} \sum_{\tau=1}^t \left(f_2(\phi(\nabla_{\boldsymbol{\theta}_{\tau-1}^1} f_1(\mathbf{x}_\tau)); \boldsymbol{\theta}^2) - (r_\tau - f_1(\mathbf{x}_\tau; \boldsymbol{\theta}_{\tau-1}^1))\right)^2$
20:     $\boldsymbol{\theta}^{2,(0)} = \boldsymbol{\theta}_0^2$
21:     **for** $k \in \{1, \dots, K_2\}$ **do**
22:         $\boldsymbol{\theta}^{2,(k)} = \boldsymbol{\theta}^{2,(k-1)} - \eta_2 \nabla_{\boldsymbol{\theta}^{2,(k-1)}} \mathcal{L}_2$
23:     **end for**
24:     $\widehat{\boldsymbol{\theta}}_t^2 = \boldsymbol{\theta}^{2,(K_2)}$
25:     Determine label $p_t$
26:     $\mathcal{L}_3 = -\frac{1}{t} \sum_{i=1}^t \left[p_t \log f_3((f_1, f_2); \boldsymbol{\theta}^3) + (1 - p_t) \log(1 - f_3((f_1, f_2); \boldsymbol{\theta}^3))\right]$
27:     $\boldsymbol{\theta}^{3,(0)} = \boldsymbol{\theta}_0^3$
28:     **for** $k \in \{1, \dots, K_3\}$ **do**
29:         $\boldsymbol{\theta}^{3,(k)} = \boldsymbol{\theta}^{3,(k-1)} - \eta_3 \nabla_{\boldsymbol{\theta}^{3,(k-1)}} \mathcal{L}_3$
30:     **end for**
31:     $\widehat{\boldsymbol{\theta}}_t^3 = \boldsymbol{\theta}^{3,(K_3)}$
32:     Randomly choose $(\boldsymbol{\theta}_t^1, \boldsymbol{\theta}_t^2)$ uniformly from $\{(\widehat{\boldsymbol{\theta}}_0^1, \widehat{\boldsymbol{\theta}}_0^2), (\widehat{\boldsymbol{\theta}}_1^1, \widehat{\boldsymbol{\theta}}_1^2), \dots, (\widehat{\boldsymbol{\theta}}_t^1, \widehat{\boldsymbol{\theta}}_t^2)\}$
33:     Randomly choose $\boldsymbol{\theta}_t^3$ uniformly from $\{\widehat{\boldsymbol{\theta}}_0^3, \widehat{\boldsymbol{\theta}}_1^3, \dots, \widehat{\boldsymbol{\theta}}_t^3\}$
34:     **Return** $\boldsymbol{\theta}_t^1, \boldsymbol{\theta}_t^2, \boldsymbol{\theta}_t^3$
35: **end procedure**

---

**3) Decision-maker**. In round $t$, given an arm $\mathbf{x}_{t,i}, i \in [n]$, with the computed exploitation score $f_1(\mathbf{x}_{t,i}; \boldsymbol{\theta}_{t-1}^1)$ and exploration score $f_2(\nabla_{\boldsymbol{\theta}_{t-1}^1} f_1; \boldsymbol{\theta}_{t-1}^2)$, we use a function $f_3\left(f_1, f_2; \boldsymbol{\theta}^3\right)$ to trade off between exploitation and exploration and compute the final score for $\mathbf{x}_{t,i}$. The selection criterion is defined as

$$\mathbf{x}_t = \arg\max_{\mathbf{x}_{t,i}, i \in [n]} f_3\left(f_1(\mathbf{x}_{t,i}; \boldsymbol{\theta}_{t-1}^1), f_2\left(\nabla_{\boldsymbol{\theta}_{t-1}^1} f_1(\mathbf{x}_{t,i}); \boldsymbol{\theta}_{t-1}^2\right); \boldsymbol{\theta}_{t-1}^3\right).$$

Note that $f_3$ can be either linear or non-linear functions. We provide the following two forms.

(1) *Linear function.* $f_3$ can be formulated as a linear function with respect to $f_1$ and $f_2$ :

$$f_3(f_1, f_2; \boldsymbol{\theta}^3) = w_1 f_1(\mathbf{x}_{t,i}; \boldsymbol{\theta}^1) + w_2 f_2(\nabla_{\boldsymbol{\theta}^1} f_1; \boldsymbol{\theta}^2)$$

where $w_1, w_2$ are two weights preset by the learner. When $w_1 = w_2 = 1$, $f_3$ can be thought of as UCB-type policy, where the estimated reward $f_1$ and potential gain $f_2$ are simply added together. In experiments, we report its empirical performance in ablation study (Appendix B).

(2) *Non-linear function*. $f_3$ also can be formulated as a neural network to learn the mapping from $(f_1, f_2)$ to the optimal arm. We transform the bandit problem into a binary classification problem. Given an arm $\mathbf{x}_{t,i}$, we define $p_{t,i}$ as the probability of being the optimal arm for $\mathbf{x}_{t,i}$ in round $t$. For brevity, we denote by $p_t$ the probability of being the optimal arm for the selected arm $\mathbf{x}_t$ in round $t$. According to different reward distributions, we have different approaches to determine $p_t$.

1. *Binary reward*. $\forall t \in [T]$, suppose $r_t$ is a binary variable over $a, b(a < b)$, it is straightforward to set: $p_t = 1.0$ if $r_t = b$; $p_t = 0.0$, otherwise.

2. *Continuous reward*. $\forall t \in [T]$, suppose $r_t$ is a continuous variable over the range $[a, b]$, we provide two ways to determine $p_t$. (1) $p_t$ can be directly set as $\frac{r_t-a}{b-a}$. (2) The learner can set a threshold $\theta, (a < \theta < b)$. Then $p_t = 1.0$ if $r_t > \theta$; $p_t = 0.0$, otherwise.

Therefore, with the collected training samples $\left\{ \left( f_1(\mathbf{x}_\tau; \boldsymbol{\theta}_{\tau-1}^1), f_2(\nabla_{\boldsymbol{\theta}_{\tau-1}^1} f_1; \boldsymbol{\theta}_{\tau-1}^2) \right), p_\tau \right\}_{\tau=1}^t$ in round $t$, we can conduct gradient descent to update parameters of $f_3(\cdot; \boldsymbol{\theta}^3)$.

Table 1 details the working structure of EE-Net. Algorithm 1 depicts the workflow of EE-Net, where the input of $f_2$ is normalized, i.e., $\phi(\nabla_{\boldsymbol{\theta}_{t-1}^1} f_1(\mathbf{x}_{t,i}))$. Algorithm 1 provides a version of gradient descent (GD) to update EE-Net, where drawing $(\boldsymbol{\theta}_t^1, \boldsymbol{\theta}_t^2)$ uniformly from their stored historical parameters is for the sake of analysis. One can easily extend EE-Net to stochastic GD to update the parameters incrementally.

**Remark 4.1** (**Network structure**). The networks $f_1, f_2, f_3$ can be different structures according to different applications. For example, in the vision tasks, $f_1$ can be set up as convolutional layers (LeCun et al., 1995). For the exploration network $f_2$, the input $\nabla_{\boldsymbol{\theta}^1} f_1$ may have exploding dimensions when the exploitation network $f_1$ becomes wide and deep, which may cause huge computation cost for $f_2$. To address this challenge, we can apply dimensionality reduction techniques to obtain low-dimensional vectors of $\nabla_{\boldsymbol{\theta}^1} f_1$. In the experiments, we use Roweis and Saul (2000) to acquire a 10-dimensional vector for $\nabla_{\boldsymbol{\theta}^1} f_1$ and achieve the best performance among all baselines. □

**Remark 4.2** (**Exploration direction**). EE-Net has the ability to determine exploration direction. Given an arm $\mathbf{x}_{t,i}$, when the estimation $f_1(\mathbf{x}_{t,i})$ is *lower* than the expected reward $h(\mathbf{x}_{t,i})$, the learner should make the "upward" exploration, i.e., increase the chance of $\mathbf{x}_{t,i}$ being explored; When $f_1(\mathbf{x}_{t,i})$ is *higher* than $h(\mathbf{x}_{t,i})$, the learner should do the "downward" exploration, i.e., decrease the chance of $\mathbf{x}_{t,i}$ being explored. EE-Net uses the neural network $f_2$ to learn $h(\mathbf{x}_{t,i}) - f_1(\mathbf{x}_{t,i})$ (which has positive and negative scores) and has the ability to determine the exploration direction. In contrast, NeuralUCB will always make "upward" exploration and NeuralTS will randomly choose between "upward" exploration and "downward" exploration (see selection criteria in Table 2 and more details in Appendix D). □

**Remark 4.3** (**Space complexity**). NeuralUCB and NeuralTS have to maintain the gradient outer product matrix (e.g., $\mathbf{A}_t = \sum_{\tau=1}^t \nabla_{\boldsymbol{\theta}^1} f_1(\mathbf{x}_\tau; \boldsymbol{\theta}_\tau^1) \nabla_{\boldsymbol{\theta}^1} f_1(\mathbf{x}_\tau; \boldsymbol{\theta}_\tau^1)^\top \in \mathbb{R}^{p \times p}$) and, for $\boldsymbol{\theta}^1 \in \mathbb{R}^p$, have a space complexity of $O(p^2)$ to store the outer product. On the contrary, EE-Net does not have this matrix and only regards $\nabla_{\boldsymbol{\theta}^1} f_1$ as the input of $f_2$. Thus, EE-Net reduces the space complexity from $\mathcal{O}(p^2)$ to $\mathcal{O}(p)$. □

## 5 REGRET ANALYSIS

In this section, we provide the regret analysis of EE-Net when $f_3$ is set as the linear function $f_3 = f_1 + f_2$, which can be thought of as the UCB-type trade-off between exploitation and exploration. For the sake of simplicity, we conduct the regret analysis on some unknown but fixed data distribution $\mathcal{D}$. In each round $t$, $n$ samples $\{(\mathbf{x}_{t,1}, r_{t,1}), (\mathbf{x}_{t,2}, r_{t,2}), \ldots, (\mathbf{x}_{t,n}, r_{t,n})\}$ are drawn i.i.d. from $\mathcal{D}$. This is standard distribution assumption in over-parameterized neural networks (Cao and Gu, 2019). Then, for the analysis, we have the following assumption, which is a standard input assumption in neural bandits and over-parameterized neural networks(Zhou et al., 2020; Allen-Zhu et al., 2019).

**Assumption 5.1** ($\rho$-Separability). For any $t \in [T], i \in [n], \|\mathbf{x}_{t,i}\|_2 = 1$, and $r_{t,i} \in [0,1]$. Then, for every pair $\mathbf{x}_{t,i}, \mathbf{x}_{t',i'}, t' \in [T], i' \in [k]$, and $(t,i) \neq (t',i')$, $\|\mathbf{x}_{t,i} - \mathbf{x}_{t',i'}\|_2 > \rho$, and suppose there exists an operator such that $\|\phi(\cdot)\|_2 = 1$ and $\|\phi(\nabla_{\boldsymbol{\theta}^1} f_1(\mathbf{x}_{t,i})) - \phi(\nabla_{\boldsymbol{\theta}^1} f_1(\mathbf{x}_{t',i'}))\|_2 \geq \rho$

For example, the operator can be designed as $\phi(\nabla_{\boldsymbol{\theta}^1} f_1(\mathbf{x}_{t,i})) = (\frac{\nabla_{\boldsymbol{\theta}^1} f_1(\mathbf{x}_{t,i})}{\sqrt{2}\|\nabla_{\boldsymbol{\theta}^1} f_1(\mathbf{x}_{t,i})\|_2}, \frac{\mathbf{x}_{t,i}}{\sqrt{2}})$. The analysis will focus on over-parameterized neural networks (Jacot et al., 2018; Du et al., 2019; Allen-Zhu et al., 2019). Given an input $\mathbf{x} \in \mathbb{R}^d$, without loss of generality, we define the fully-connected network $f$ with depth $L \geq 2$ and width $m$:

$$f(\mathbf{x}; \boldsymbol{\theta}) = \mathbf{W}_L \sigma(\mathbf{W}_{L-1}\sigma(\mathbf{W}_{L-2}\ldots\sigma(\mathbf{W}_1\mathbf{x}))) \tag{5.1}$$

where $\sigma$ is the ReLU activation function, $\mathbf{W}_1 \in \mathbb{R}^{m \times d}$, $\mathbf{W}_l \in \mathbb{R}^{m \times m}$, for $2 \leq l \leq L - 1$, $\mathbf{W}^L \in \mathbb{R}^{1 \times m}$, and $\boldsymbol{\theta} = [\text{vec}(\mathbf{W}_1)^\mathsf{T}, \text{vec}(\mathbf{W}_2)^\mathsf{T}, \ldots, \text{vec}(\mathbf{W}_L)^\mathsf{T}]^\mathsf{T}$.

*Initialization.* For any $l \in [L-1]$, each entry of $\mathbf{W}_l$ is drawn from the normal distribution $\mathcal{N}(0, \frac{2}{m})$ and $\mathbf{W}_L$ is drawn from the normal distribution $\mathcal{N}(0, \frac{1}{m})$. Note that EE-Net at most has three networks $f_1, f_2, f_3$. We define them following the definition of $f$ for brevity, although they may have different depth or width. Then, we have the following theorem for EE-Net. Recall that $\eta_1, \eta_2$ are the learning rates for $f_1, f_2$; $K_1$ is the number of iterations of gradient descent for $f_1$ in each round; and $K_2$ is the number of iterations for $f_2$.

**Theorem 1.** *Let $f_1, f_2$ follow the setting of $f$ (Eq. (5.1)) with the same width $m$ and depth $L$. Let $\mathcal{L}_1, \mathcal{L}_2$ be loss functions defined in Algorithm 1. Set $f_3$ as $f_3 = f_1 + f_2$. For any $\delta \in (0,1), \epsilon \in (0, \mathcal{O}(\frac{1}{T})], \rho \in (0, \mathcal{O}(\frac{1}{L})]$, suppose*

$$m \geq \widetilde{\Omega}\left(poly(T, n, L, \rho^{-1}) \cdot \log(1/\delta) \cdot e^{\sqrt{\log(Tn/\delta)}}\right),$$

$$\eta_1 = \eta_2 = \min\left(\Theta\left(\frac{T^5}{\sqrt{2}\delta^2 m}\right), \Theta\left(\frac{\rho}{poly(T, n, L) \cdot m}\right)\right), \tag{5.2}$$

$$K_1 = K_2 = \Theta\left(\frac{poly(T, n, L)}{\rho\delta^2} \cdot \log\left(\epsilon^{-1}\right)\right).$$

*Then, with probability at least $1 - \delta$ over the initialization, the pseudo regret of EE-Net in $T$ rounds satisfies*

$$\mathbf{R}_T \leq \mathcal{O}(1) + (2\sqrt{T} - 1)3\sqrt{2}\mathcal{O}(L) + \mathcal{O}\left((2\sqrt{T} - 1)\sqrt{2\log\frac{\mathcal{O}(Tn)}{\delta}}\right). \tag{5.3}$$

**Comparison with existing works**. Under the similar assumptions in over-parameterized neural networks, the regret bounds complexity of NeuralUCB (Zhou et al., 2020) and NeuralTS (Zhang et al., 2021) both are

$$\mathbf{R}_T \leq \mathcal{O}\left(\sqrt{\tilde{d}T \log T}\right) \cdot \mathcal{O}\left(\sqrt{\tilde{d}\log T}\right), \text{ and } \tilde{d} = \frac{\log\det(\mathbf{I} + \mathbf{H}/\lambda)}{\log(1 + Tn/\lambda)}$$

where $\mathbf{H}$ is the neural tangent kernel matrix (NTK) (Jacot et al., 2018; Arora et al., 2019) and $\lambda$ is a regularization parameter. Similarly, in linear contextual bandits, Abbasi-Yadkori et al. (2011) achieve $\mathcal{O}(d\sqrt{T}\log T)$ and Li et al. (2017) achieve $\mathcal{O}(\sqrt{dT}\log T)$.

**Remark 5.1.** Compared to NeuralUCB/TS, EE-Net roughly improves by a multiplicative factor of $\sqrt{\log T}$, because our proof of EE-Net is directly built on recent advances in convergence theory (Allen-Zhu et al., 2019) and generalization bound (Cao and Gu, 2019) of over-parameterized neural networks. Instead, the analysis for NeuralUCB/TS contains three parts of approximation error by calculating the distances between the expected reward and ridge regression, ridge regression and NTK, and NTK and network function. □

**Remark 5.2.** The regret bound of EE-Net does not have the effective dimension $\tilde{d}$ or input dimension $d$. $\tilde{d}$ or $d$ may cause significant error, when the determinant of $\mathbf{H}$ is extremely large or $d > T$. □

The proof of Theorem 1 is in Appendix C and mainly based on the following generalization bound. The bound results from an online-to-batch conversion while using convergence guarantees of deep learning optimization.

**Lemma 5.1.** *For any $\delta \in (0,1), \epsilon \in (0,1), \rho \in (0, \mathcal{O}(\frac{1}{L}))$, suppose $m, \eta_1, \eta_2, K_1, K_2$ satisfy the conditions in Eq. (5.2) and $(\mathbf{x}_{\tau,i}, r_{\tau,i}) \sim \mathcal{D}, \forall \tau \in [t], i \in [n]$. Let*

$$\mathbf{x}_t = \arg \max_{\mathbf{x}_{t,i}, i \in [n]} \left[ f_2 \left( \phi(\nabla_{\boldsymbol{\theta}_{t-1}^1} f_1(\mathbf{x}_{t,i}; \boldsymbol{\theta}_{t-1}^1)); \boldsymbol{\theta}_{t-1}^2 \right) + f_1(\mathbf{x}_{t,i}; \boldsymbol{\theta}_{t-1}^1) \right],$$

*and $r_t$ is the corresponding reward, given $(\mathbf{x}_{t,i}, r_{t,i}), i \in [n]$. Then, with probability at least $(1 - \delta)$ over the random of the initialization, it holds that*

$$\begin{aligned}
\mathbb{E}_{(\mathbf{x}_{t,i}, r_{t,i}), i \in [n]} & \left[ \left| f_2 \left( \phi(\nabla_{\boldsymbol{\theta}_{t-1}^1} f_1(\mathbf{x}_{t,i}; \boldsymbol{\theta}_{t-1}^1)); \boldsymbol{\theta}_{t-1}^2 \right) - (r_t - f_1(\mathbf{x}_t; \boldsymbol{\theta}_{t-1}^1)) \right| \mid \{\mathbf{x}_\tau, r_\tau\}_{\tau=1}^{t-1} \right] \\
& \leq \sqrt{\frac{2\epsilon}{t}} + \mathcal{O}\left(\frac{3L}{\sqrt{2t}}\right) + (1 + 2\xi)\sqrt{\frac{2\log(\mathcal{O}(tn/\delta))}{t}},
\end{aligned}$$
(5.4)

*where the expectation is also taken over $(\boldsymbol{\theta}_{t-1}^1, \boldsymbol{\theta}_{t-1}^2)$ that are uniformly drawn from $(\widehat{\boldsymbol{\theta}}_\tau^1, \widehat{\boldsymbol{\theta}}_\tau^2), \tau \in [t-1]$.*

**Remark 5.3.** Lemma 5.1 provides a fixed $\tilde{\mathcal{O}}(\frac{1}{\sqrt{t}})$-rate generalization bound for exploitation-exploration networks $f_1, f_2$ in contrast with the relative bound w.r.t. the Neural Tangent Random Feature (NTRF) benchmark (Cao and Gu, 2019). We achieve this by working in the regression rather than classification setting and utilizing the convergence guarantees for square loss (Allen-Zhu et al., 2019). Note that the bound in Lemma 5.1 holds in the setting of bounded (possibly random) rewards $r \in [0,1]$ instead of a fixed function in the conventional classification setting.

## 6 EXPERIMENTS

In this section, we evaluate EE-Net on four real-world datasets comparing with strong state-of-the-art baselines. We first present the setup of experiments, then show regret comparison and report ablation study. Codes are available at [1].

We use four real-world datasets: **Mnist, Yelp, Movielens, and Disin**, the details and settings of which are attached in Appendix A.

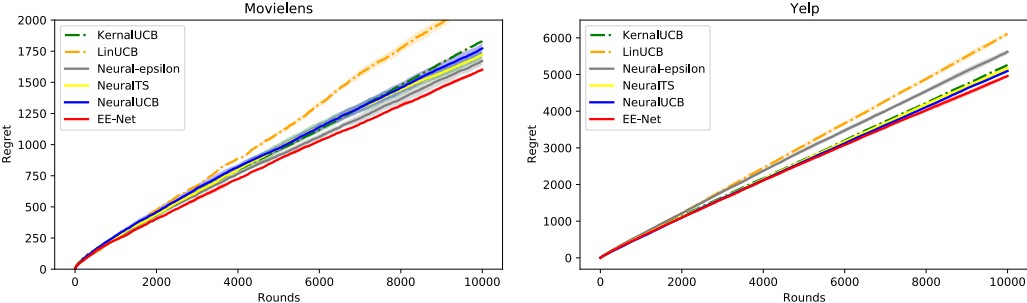

Figure 2: Regret comparison on Movielens and Yelp (mean of 10 runs with standard deviation (shadow)). With the same exploitation network $f_1$, EE-Net outperforms all baselines.

**Baselines**. To comprehensively evaluate EE-Net, we choose 3 neural-based bandit algorithms, one linear and one kernelized bandit algorithms.

1. LinUCB (Li et al., 2010) explicitly assumes the reward is a linear function of arm vector and unknown user parameter and then applies the ridge regression and un upper confidence bound to determine selected arm.

2. KernelUCB (Valko et al., 2013) adopts a predefined kernel matrix on the reward space combined with a UCB-based exploration strategy.

3. Neural-Epsilon adapts the epsilon-greedy exploration strategy on exploitation network $f_1$. I.e., with probability $1 - \epsilon$, the arm is selected by $\mathbf{x}_t = \arg \max_{i \in [n]} f_1(\mathbf{x}_{t,i}; \boldsymbol{\theta}^1)$ and with probability $\epsilon$, the arm is chosen randomly.

---

[1]https://github.com/banyikun/EE-Net-ICLR-2022

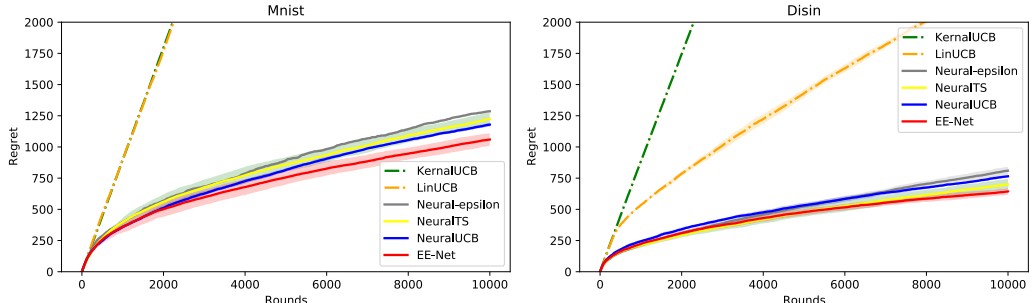

Figure 3: Regret comparison on Mnist and Disin (mean of 10 runs with standard deviation (shadow)). With the same exploitation network $f_1$, EE-Net outperforms all baselines.

4. NeuralUCB (Zhou et al., 2020) uses the exploitation network $f_1$ to learn the reward function coming with an UCB-based exploration strategy.

5. NeuralTS (Zhang et al., 2021) adopts the exploitation network $f_1$ to learn the reward function coming with an Thompson Sampling exploration strategy.

Note that we do not report results of LinTS and KernelTS in experiments, because of the limited space in figures, but LinTS and KernelTS have been significantly outperformed by NeuralTS (Zhang et al., 2021).

**Setup for EE-Net**. To compare fairly, for all the neural-based methods including EE-Net, the exploitation network $f_1$ is built by a 2-layer fully-connected network with 100 width. For the exploration network $f_2$, we use a 2-layer fully-connected network with 100 width as well. For the decision maker $f_3$, by comprehensively evaluate both linear and nonlinear functions, we found that the most effective approach is combining them together, which we call " *hybrid decision maker*". In detail, for rounds $t \leq 500$, $f_3$ is set as $f_3 = f_2 + f_1$, and for $t > 500$, $f_3$ is set as a neural network with two 20-width fully-connected layers. Setting $f_3$ in this way is because the linear decision maker can maintain stable performance in each running (robustness) and the non-linear decision maker can further improve the performance (see details in Appendix B). The hybrid decision maker can combine these two advantages together. The configurations of all methods are attached in Appendix A.

**Results**. Figure 2 and Figure 3 show the regret comparison on these four datasets. EE-Net consistently outperforms all baselines across all datasets. For LinUCB and KernelUCN, the simple linear reward function or predefined kernel cannot properly formulate ground-truth reward function existed in real-world datasets. In particular, on Mnist and Disin datasets, the correlations between rewards and arm feature vectors are not linear or some simple mappings. Thus, LinUCB and KernelUCB barely exploit the past collected data samples and fail to select correct arms. For neural-based bandit algorithms, the exploration probability of Neural-Epsilon is fixed and difficult to be adjustable. Thus it is usually hard to make effective exploration. To make exploration, NeuralUCB statistically calculates a gradient-based upper confidence bound and NeuralTS draws each arm's predicted reward from a normal distribution where the standard deviation is computed by gradient. However, the confidence bound or standard deviation they calculated only consider the worst cases and thus may not be able represent the actual potential of each arm, and they cannot make "upward" and "downward" exploration properly. Instead, EE-Net uses a neural network $f_2$ to learn each arm's potential by neural network's powerful representation ability. Therefore, EE-Net can outperform these two state-of-the-art bandit algorithms. Note that NeuralUCB/TS does need two parameters to tune UCB/TS according to different scenarios while EE-Net only needs to set up a neural network and automatically learns it.

**Ablation Study**. In Appendix B, we conduct ablation study regarding the label function $y$ of $f_2$ and the different setting of $f_3$.

## 7 CONCLUSION

In this paper, we propose a novel exploration strategy, EE-Net. In addition to a neural network that exploits collected data in past rounds , EE-Net has another neural network to learn the potential gain compared to current estimation for exploration. Then, a decision maker is built to make selections to further trade off between exploitation and exploration. We demonstrate that EE-Net outperforms NeuralUCB and NeuralTS both theoretically and empirically, becoming the new state-of-the-art exploration policy.

**Acknowledgements:** We are grateful to Shiliang Zuo and Yunzhe Qi for the valuable discussions in the revisions of EE-Net. This research work is supported by National Science Foundation under Awards No. IIS-1947203, IIS-2002540, IIS-2137468, IIS-1908104, OAC-1934634, and DBI-2021898, and a grant from C3.ai. The views and conclusions are those of the authors and should not be interpreted as representing the official policies of the funding agencies or the government.

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

# A   DATASETS AND SETUP

**MNIST dataset**.  MNIST is a well-known image dataset (LeCun et al., 1998) for the 10-class classification problem. Following the evaluation setting of existing works (Valko et al., 2013; Zhou et al., 2020; Zhang et al., 2021), we transform this classification problem into bandit problem. Consider an image $\mathbf{x} \in \mathbb{R}^d$, we aim to classify it from 10 classes. First, in each round, the image $\mathbf{x}$ is transformed into 10 arms and presented to the learner, represented by 10 vectors in sequence $\mathbf{x}_1 = (\mathbf{x}, \mathbf{0}, \dots, \mathbf{0}), \mathbf{x}_2 = (\mathbf{0}, \mathbf{x}, \dots, \mathbf{0}), \dots, \mathbf{x}_{10} = (\mathbf{0}, \mathbf{0}, \dots, \mathbf{x}) \in \mathbb{R}^{10d}$. The reward is defined as 1 if the index of selected arm matches the index of $\mathbf{x}$'s ground-truth class; Otherwise, the reward is 0.

**Yelp[2] and Movielens (Harper and Konstan, 2015) datasets**. Yelp is a dataset released in the Yelp dataset challenge, which consists of 4.7 million rating entries for $1.57 \times 10^5$ restaurants by 1.18 million users. MovieLens is a dataset consisting of 25 million ratings between $1.6 \times 10^5$ users and $6 \times 10^4$ movies. We build the rating matrix by choosing the top 2000 users and top 10000 restaurants(movies) and use singular-value decomposition (SVD) to extract a 10-dimension feature vector for each user and restaurant(movie). In these two datasets, the bandit algorithm is to choose the restaurants(movies) with bad ratings. We generate the reward by using the restaurant(movie)'s gained stars scored by the users. In each rating record, if the user scores a restaurant(movie) less than 2 stars (5 stars totally), its reward is 1; Otherwise, its reward is 0. In each round, we set 10 arms as follows: we randomly choose one with reward 1 and randomly pick the other 9 restaurants(movies) with 0 rewards; then, the representation of each arm is the concatenation of corresponding user feature vector and restaurant(movie) feature vector.

**Disin (Ahmed et al., 2018) dataset**. Disin is a fake news dataset on kaggle[3] including 12600 fake news articles and 12600 truthful news articles, where each article is represented by the text. To transform the text into vectors, we use the approach (Fu and He, 2021) to represent each article by a 300-dimension vector. Similarly, we form a 10-arm pool in each round, where 9 real news and 1 fake news are randomly selected. If the fake news is selected, the reward is 1; Otherwise, the reward is 0.

**Configurations**. For LinUCB, following (Li et al., 2010), we do a grid search for the exploration constant $\alpha$ over $(0.01, 0.1, 1)$ which is to tune the scale of UCB. For KernelUCB (Valko et al., 2013), we use the radial basis function kernel and stop adding contexts after 1000 rounds, following (Valko et al., 2013; Zhou et al., 2020). For the regularization parameter $\lambda$ and exploration parameter $\nu$ in KernelUCB, we do the grid search for $\lambda$ over $(0.1, 1, 10)$ and for $\nu$ over $(0.01, 0.1, 1)$. For NeuralUCB and NeuralTS, following setting of (Zhou et al., 2020; Zhang et al., 2021), we use the exploiation network $f_1$ and conduct the grid search for the exploration parameter $\nu$ over $(0.001, 0.01, 0.1, 1)$ and for the regularization parameter $\lambda$ over $(0.01, 0.1, 1)$. For NeuralEpsilon, we use the same neural network $f_1$ and do the grid search for the exploration probability $\epsilon$ over $(0.01, 0.1, 0.2)$. For the neural bandits NeuralUCB/TS, following their setting, as they have expensive computation cost to store and compute the whole gradient matrix, we use a diagonal matrix to make approximation. For all neural networks, we conduct the grid search for learning rate over $(0.01, 0.001, 0.0005, 0.0001)$. For all grid-searched parameters, we choose the best of them for the comparison and report the averaged results of 10 runs for all methods.

**Create exploration samples for $f_2$.** When the selected arm is not optimal in a round, the optimal arm must exist among the remaining arms, and thus the exploration consideration should be added to the remaining arms. Based on this fact, we create additional samples for the exploration network $f_2$ in practice. For example, in setting of binary reward, e.g., 0 or 1 reward, if the received reward $r_t = 0$ while select $\mathbf{x}_t$, we add new train samples for $f_2$, $(\mathbf{x}_{t,i}, c_r)$ for each $i \in [i] \cap \mathbf{x}_{t,i} \neq \mathbf{x}_t$, where $c_r \in (0, 1)$ usually is a small constant. This measure can further improve the performance of EE-Net in our experiments.

# B   ABLATION STUDY

In this section, we conduct ablation study regarding the label function $y$ for exploration network $f_2$ and seting of decision maker $f_3$ on two representative datasets Movielens and Mnist.

---

[2]https://www.yelp.com/dataset
[3]https://www.kaggle.com/clmentbisaillon/fake-and-real-news-dataset

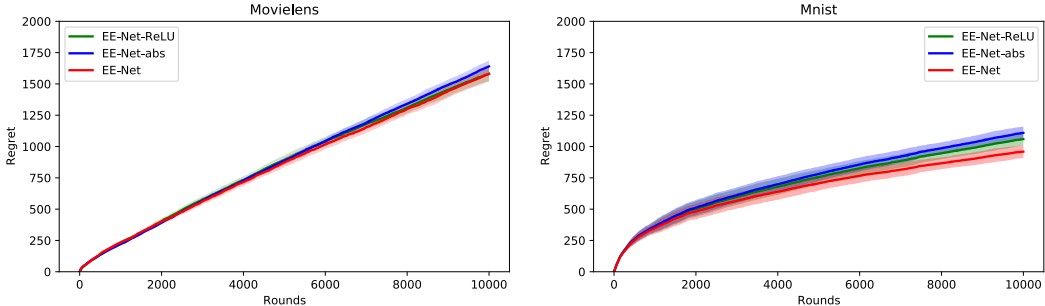

Figure 4: Ablation study on label function $y$ for $f_2$. EE-Net denotes $y_1 = r - f_1$, EE-Net-abs denotes $y_2 = |r - f_1|$, and EE-Net-ReLU denotes $y_3 = \text{ReLU}(r - f_1)$. EE-Net shows the best performance on these two datasets.

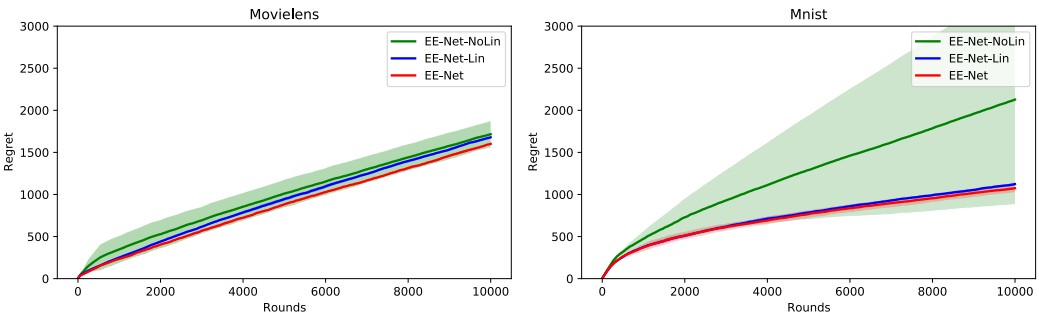

Figure 5: Ablation study on decision maker $f_3$. EE-Net-Lin denotes $f_3 = f_1 + f_2$, EE-Net-NoLin denote the nonlinear one where $f_3$ is a neural network (2 layer, width 20), EE-Net denotes the hybrid one where $f_3 = f_1 + f_2$ if $t \leq 500$ and $f_3$ is the neural network if $t > 500$. EE-Net has the most stable and best performance.

**Label function** $y$. In this paper, we use $y_1 = r - f_1$ to measure the potential gain of an arm, as the label of $f_2$. Moreover, we provide other two intuitive form $y_2 = |r - f_1|$ and $y_3 = ReLU(r - f_1)$. Figure 4 shows the regret with different $y$, where "EE-Net" denotes our method with default $y_1$, "EE-Net-abs" represents the one with $y_2$ and "EE-Net-ReLU" is with $y_3$. On Movielens and Mnist datasets, EE-Net slightly outperforms EE-Net-abs and EE-Net-ReLU. In fact, $y_1$ can effectively represent the positive potential gain and negative potential gain, such that $f_2$ intends to score the arm with positive gain higher and score the arm with negative gain lower. However, $y_2$ treats the positive/negative potential gain evenly, weakening the discriminative ability. $y_3$ can recognize the positive gain while neglecting the difference of negative gain. Therefore, $y_1$ usually is the most effective one for empirical performance.

**Setting of** $f_3$. $f_3$ can be set as an either linear function or non-linear function. In the experiment, we test the simple linear function $f_3 = f_1 + f_2$, denoted by "EE-Net-Lin", and a non-linear function represented by a 2-layer 20-width fully-connected neural network, denoted by "EE-Net-NoLin". For the default hybrid setting, denoted by "EE-Net", when rounds $t \leq 500$, $f_3 = f_1 + f_2$; Otherwise, $f_3$ is the neural network. Figure 5 reports the regret with these three different modes. EE-Net achieves the best performance with small standard deviation. In contrast, EE-Net-NoLin obtains the worst performance and largest standard deviation. However, notice that EE-Net-NoLin can achieve the best performance in certain running (the green shallow) but it is erratic. Because in the begin phase, without enough training samples, EE-Net-NoLin strongly relies on the quality of collected samples. With appropriate training samples, gradient descent can lead $f_3$ to global optimum. On the other hand, with misleading training samples, gradient descent can deviate $f_3$ from global optimum. Therefore, EE-Net-NoLin shows very unstable performance. In contrast, EE-Net-Lin is inspired by the UCB strategy, i.e., the exploitation plus the exploration, exhibiting stable performance. To combine their

## C  PROOF OF THEOREM 1

In this section, we provide the proof of Theorem 1 and related lemmas.

*Proof.* For brevity, for the selected arm $\mathbf{x}_t$ in round $t$, let $h(\mathbf{x}_t)$ be its expected reward and $\mathbf{x}_t^* = \arg\max_{\mathbf{x}_{t,i}, i \in [n]} h(\mathbf{x}_{t,i})$ be the optimal arm in round $t$. Let $f_3(\mathbf{x}; \boldsymbol{\theta}_{t-1}) = f_2\left(\phi(\nabla_{\boldsymbol{\theta}_{t-1}^1} f_1(\mathbf{x}; \boldsymbol{\theta}_{t-1}^1)); \boldsymbol{\theta}_{t-1}^2\right) + f_1(\mathbf{x}; \boldsymbol{\theta}_{t-1}^1)$.

Note that $(\mathbf{x}_{t,i}, r_{t,i}) \sim \mathcal{D}$, for each $i \in [n]$. Then, the expected regret of round $t$ is given by

$$
\begin{aligned}
R_t &= \mathbb{E}_{\mathbf{x}_{t,i}, i \in [n]}[h(\mathbf{x}_t^*) - h(\mathbf{x}_t)] \\
&= \mathbb{E}_{\mathbf{x}_{t,i}, i \in [n]}[h(\mathbf{x}_t^*) - f_3(\mathbf{x}_t) + f_3(\mathbf{x}_t) - h(\mathbf{x}_t)] \\
&\leq \mathbb{E}_{\mathbf{x}_{t,i}, i \in [n]}[\underbrace{h(\mathbf{x}_t^*) - f_3(\mathbf{x}_t^*) + f_3(\mathbf{x}_t) - f_3(\mathbf{x}_t)}_{I_1} + f_3(\mathbf{x}_t) - h(\mathbf{x}_t)] \\
&= \mathbb{E}_{\mathbf{x}_{t,i}, i \in [n]}[h(\mathbf{x}_t^*) - f_3(\mathbf{x}_t^*) + f_3(\mathbf{x}_t) - h(\mathbf{x}_t)] \\
&= \mathbb{E}_{\mathbf{x}_{t,i}, i \in [n]}[h(\mathbf{x}_t^*) - f_3(\mathbf{x}_t^*; \boldsymbol{\theta}_{t-1}) + f_3(\mathbf{x}_t; \boldsymbol{\theta}_{t-1}) - h(\mathbf{x}_t)] \\
&\overset{(a)}{=} \mathbb{E}_{\mathbf{x}_{t,i}, i \in [n]}[h(\mathbf{x}_t^*) - f_3(\mathbf{x}_t^*; \boldsymbol{\theta}_{t-1}^*) + f_3(\mathbf{x}_t^*; \boldsymbol{\theta}_{t-1}^*) - f_3(\mathbf{x}_t^*; \boldsymbol{\theta}_{t-1}) + f_3(\mathbf{x}_t; \boldsymbol{\theta}_{t-1}) - h(\mathbf{x}_t)] \\
&= \mathbb{E}_{\mathbf{x}_{t,i}, i \in [n]}[h(\mathbf{x}_t^*) - f_3(\mathbf{x}_t^*; \boldsymbol{\theta}_{t-1}^*)]] + \mathbb{E}_{\mathbf{x}_{t,i}, i \in [n]}[f_3(\mathbf{x}_t^*; \boldsymbol{\theta}_{t-1}^*) - f_3(\mathbf{x}_t^*; \boldsymbol{\theta}_{t-1})] \\
&\quad + \mathbb{E}_{\mathbf{x}_{t,i}, i \in [n]}[f_3(\mathbf{x}_t; \boldsymbol{\theta}_{t-1}) - h(\mathbf{x}_t)] \\
&\leq \underbrace{\mathbb{E}_{(\mathbf{x}_{t,i}, r_{t,i}), i \in [n]}\left[\left|f_2\left(\phi(\nabla_{\boldsymbol{\theta}_{t-1}^{1,*}} f_1(\mathbf{x}_t^*; \boldsymbol{\theta}_{t-1}^{1,*})); \boldsymbol{\theta}_{t-1}^{2,*}\right) - \left(r_t^* - f_1(\mathbf{x}_t^*; \boldsymbol{\theta}_{t-1}^{1,*})\right)\right|\right]}_{I_2} \\
&\quad + \underbrace{\mathbb{E}_{\mathbf{x}_{t,i}, i \in [n]}\left[\left|f_2\left(\phi(\nabla_{\boldsymbol{\theta}_{t-1}^{1,*}} f_1(\mathbf{x}_t^*; \boldsymbol{\theta}_{t-1}^{1,*})); \boldsymbol{\theta}_{t-1}^{2,*}\right) - f_2\left(\phi(\nabla_{\boldsymbol{\theta}_{t-1}^1} f_1(\mathbf{x}_t^*; \boldsymbol{\theta}_{t-1}^1)); \boldsymbol{\theta}_{t-1}^2\right)\right|\right]}_{I_3} \\
&\quad + \underbrace{\mathbb{E}_{\mathbf{x}_{t,i}, i \in [n]}\left[\left|f_1(\mathbf{x}_t^*; \boldsymbol{\theta}_{t-1}^{1,*}) - f_1(\mathbf{x}_t^*; \boldsymbol{\theta}_{t-1}^1)\right|\right]}_{I_4} \\
&\quad + \underbrace{\mathbb{E}_{(\mathbf{x}_{t,i}, r_{t,i}), i \in [n]}\left[\left|f_2\left(\phi(\nabla_{\boldsymbol{\theta}_{t-1}^1} f_1(\mathbf{x}_t; \boldsymbol{\theta}_{t-1}^1)); \boldsymbol{\theta}_{t-1}^2\right) - \left(r_t - f_1(\mathbf{x}_t; \boldsymbol{\theta}_{t-1}^1)\right)\right|\right]}_{I_5}
\end{aligned}
$$

$$\text{(C.1)}$$

where $I_1$ is because $f_3(\mathbf{x}_t) = \max_{i \in [n]} f_3(\mathbf{x}_{t,i})$ and $f_3(\mathbf{x}_t) - f_3(\mathbf{x}_t^*) \geq 0$ and $(a)$ introduces the additional parameters $\boldsymbol{\theta}_{t-1}^* = (\boldsymbol{\theta}_{t-1}^{1,*}, \boldsymbol{\theta}_{t-1}^{2,*})$ which will be suitably chosen.

Because, for each $i \in [n]$, $(\mathbf{x}_{t,i}, r_{t,i}) \sim \mathcal{D}$, applying Lemma C.1 and Corollary C.1, with probability at least $(1 - \delta)$ over the randomness of initialization, for $I_2, I_5$, we have

$$
I_2, I_5 \leq \sqrt{\frac{2\epsilon}{t}} + \mathcal{O}\left(\frac{3L}{\sqrt{2t}}\right) + (1 + 2\xi)\sqrt{\frac{2\log(\mathcal{O}(tn)/\delta)}{t}}, \tag{C.2}
$$

where

$$
\xi = \mathcal{O}(1) + \mathcal{O}\left(\frac{t^3 n L \log m}{\rho \sqrt{m}}\right) + \mathcal{O}\left(\frac{t^4 n L^2 \log^{11/6} m}{\rho^{4/3} m^{1/6}}\right) \tag{C.3}
$$

and we apply the union bound over round $\tau, \forall \tau \in [t]$ to make Lemma C.1 and Corollary C.1 hold for each round $\tau, \tau \in [t]$.

For $I_3, I_4$, based on Lemma C.2, with probability at least $1 - \delta$, we have

$$
I_3, I_4 \leq \left(1 + \mathcal{O}\left(\frac{t L^3 \log^{5/6} m}{\rho^{1/3} m^{1/6}}\right)\right) \mathcal{O}\left(\frac{L t^3}{\rho \sqrt{m}} \log m\right) + \mathcal{O}\left(\frac{t^4 L^2 \log^{11/6} m}{\rho^{4/3} m^{1/6}}\right) := \xi_1. \tag{C.4}
$$

To sum up, with probability at least $1 - \delta$, we have

$$R_t \leq 2\left(\sqrt{\frac{2\epsilon}{t}} + \mathcal{O}\left(\frac{3L}{\sqrt{2t}}\right) + (1 + 2\xi)\sqrt{\frac{2\log(\mathcal{O}(tn)/\delta)}{t}} + \xi_1\right). \tag{C.5}$$

Then expected regret of $T$ rounds is computed by

$$\mathbf{R}_T = \sum_{t=1}^{T} R_t$$

$$\leq 2\sum_{t=1}^{T}\left(\sqrt{\frac{2\epsilon}{t}} + \mathcal{O}\left(\frac{3L}{\sqrt{2t}}\right) + (1 + 2\xi)\sqrt{\frac{2\log(\mathcal{O}(tn)/\delta)}{t}} + \xi_1\right)$$

$$\leq \underbrace{(2\sqrt{T} - 1)2\sqrt{2\epsilon} + (2\sqrt{T} - 1)3\sqrt{2}\mathcal{O}(L) + 2(1 + 2\xi)(2\sqrt{T} - 1)\sqrt{2\log(\mathcal{O}(Tn)/\delta)}}_{I_2} + \mathcal{O}(1)$$

$$= (2\sqrt{T} - 1)(2\sqrt{2\epsilon} + 3\sqrt{2}\mathcal{O}(L)) + 2(1 + 2\xi)(2\sqrt{T} - 1)\sqrt{2\log(\mathcal{O}(Tn)/\delta)} + \mathcal{O}(1) \tag{C.6}$$

where $I_2$ is because $\sum_{t=1}^{T}\frac{1}{\sqrt{t}} \leq \int_1^T \frac{1}{\sqrt{t}}\,dx + 1 = 2\sqrt{T} - 1$ (Chlebus, 2009) and the bound of $\xi_1$ is due to the choice of $m$, i.e., since $\xi_1 = \widetilde{\mathcal{O}}(1/m^{1/6})$ and $m \geq \widetilde{\Omega}(\text{poly}(T))$, $m$ can be chosen so that $T\xi_1 = \widetilde{\mathcal{O}}(T/m^{1/6}) \leq \mathcal{O}(1)$.

Then, when $\epsilon \leq 1/T$, we have

$$\mathbf{R}_T \leq \mathcal{O}(1) + (2\sqrt{T} - 1)3\sqrt{2}\mathcal{O}(L) + 2(1 + 2\xi)(2\sqrt{T} - 1)\sqrt{2\log(\mathcal{O}(Tn)/\delta)}. \tag{C.7}$$

As the choice of $m$, we have $\xi \leq \mathcal{O}(1)$. Therefore, we have

$$\mathbf{R}_T \leq \mathcal{O}(1) + (2\sqrt{T} - 1)3\sqrt{2}\mathcal{O}(L) + \mathcal{O}\left((2\sqrt{T} - 1)\sqrt{2\log(\mathcal{O}(Tn)/\delta)}\right). \tag{C.8}$$

The proof is completed. $\qquad\square$

**Lemma C.1.** *[**Lemma 5.1 restated**] For any $\delta, \epsilon \in (0, 1), \rho \in (0, \mathcal{O}(\frac{1}{L}))$, suppose $m, \eta_1, \eta_2, K_1, K_2$ satisfy the conditions in Eq. (5.2) and $(\mathbf{x}_{\tau,i}, r_{\tau,i}) \sim \mathcal{D}, \forall \tau \in [t], i \in [n]$. Let*

$$\mathbf{x}_t = \arg\max_{\mathbf{x}_{t,i}, i \in [n]}\left[f_2\left(\phi(\nabla_{\boldsymbol{\theta}_{t-1}^1}f_1(\mathbf{x}_{t,i}; \boldsymbol{\theta}_{t-1}^1)); \boldsymbol{\theta}_{t-1}^2\right) + f_1(\mathbf{x}_{t,i}; \boldsymbol{\theta}_{t-1}^1)\right],$$

*and $r_t$ is the corresponding reward, given $(\mathbf{x}_{t,i}, r_{t,i}), i \in [n]$. Then, with probability at least $(1 - \delta)$ over the random of the initialization, it holds that*

$$\mathop{\mathbb{E}}_{(\mathbf{x}_{t,i}, r_{t,i}), i \in [n]}\left[\left|f_2\left(\phi(\nabla_{\boldsymbol{\theta}_{t-1}^1}f_1(\mathbf{x}_t; \boldsymbol{\theta}_{t-1}^1)); \boldsymbol{\theta}_{t-1}^2\right) - (r_t - f_1(\mathbf{x}_t; \boldsymbol{\theta}_{t-1}^1))\right| \mid \{\mathbf{x}_\tau, r_\tau\}_{\tau=1}^{t-1}\right]$$

$$\leq \sqrt{\frac{2\epsilon}{t}} + \mathcal{O}\left(\frac{3L}{\sqrt{2t}}\right) + (1 + 2\xi)\sqrt{\frac{2\log(\mathcal{O}(tn/\delta))}{t}}, \tag{C.9}$$

*where the expectation is also taken over $(\boldsymbol{\theta}_{t-1}^1, \boldsymbol{\theta}_{t-1}^2)$ that are uniformly drawn from $(\widehat{\boldsymbol{\theta}}_\tau^1, \widehat{\boldsymbol{\theta}}_\tau^2), \tau \in [t-1]$.*

*Proof.* In this proof, we consider the collected data of up to round $t - 1$, $\{\mathbf{x}_\tau, r_\tau\}_{\tau=1}^{t-1}$, as the training dataset and then obtain a generalization bound for it, inspired by Cao and Gu (2019).

For convenience, we use $\mathbf{x} := \mathbf{x}_{t,i}, r := r_{t,i}$, noting that the same analysis holds for each $i \in [n]$. Consider the exploration network $f_2$, applying Lemma C.3. With probability at least $1 - \delta$, for any $\tau \in [t]$, we have

$$\left|f_2\left(\phi(\nabla_{\widehat{\boldsymbol{\theta}}_\tau^1}f_1(\mathbf{x}; \widehat{\boldsymbol{\theta}}_\tau^1)); \widehat{\boldsymbol{\theta}}_\tau^2\right)\right| \leq \xi. \tag{C.10}$$

Similarly, applying Lemma C.3 again, with probability at least $1 - \delta$, for any $t \in [T]$, we have

$$|f_1(\mathbf{x}; \widehat{\boldsymbol{\theta}}_\tau^1)| \leq \xi \tag{C.11}$$

Because for any $r \sim \mathcal{D}_r$, $|r| \leq 1$, with Eq. (C.10) and (C.11), applying union bound, with probability at least $(1 - 2\delta)$ over the random initialization, we have

$$\left| f_2 \left( \phi(\nabla_{\widehat{\boldsymbol{\theta}}_\tau^1} f_1(\mathbf{x}; \widehat{\boldsymbol{\theta}}_\tau^1)); \widehat{\boldsymbol{\theta}}_\tau^2 \right) - (r - f_1(\mathbf{x}; \widehat{\boldsymbol{\theta}}_\tau^1)) \right| \leq 1 + 2\xi. \tag{C.12}$$

Noting that (C.12) is for $\mathbf{x} = \mathbf{x}_{\tau,i}$ for a specific $\tau \in [t], i \in [n]$. By union bound, (C.12) holds $\forall \tau \in [t], i \in [n]$ with probability at least $(1 - nt\delta)$. For brevity, let $f_2(\mathbf{x}; \widehat{\boldsymbol{\theta}}_\tau^2)$ represent $f_2 \left( \phi(\nabla_{\widehat{\boldsymbol{\theta}}_\tau^1} f_1(\mathbf{x}; \widehat{\boldsymbol{\theta}}_\tau^1)); \widehat{\boldsymbol{\theta}}_\tau^2 \right)$.

Recall that, for each $\tau \in [t-1]$, $\widehat{\boldsymbol{\theta}}_\tau^1$ and $\widehat{\boldsymbol{\theta}}_\tau^2$ are the parameters training on $\{\mathbf{x}_{\tau'}, r_{\tau'}\}_{\tau'=1}^\tau$ according to Algorithm 1. In round $\tau \in [t]$, let $\mathbf{x}_\tau = \arg\max_{\mathbf{x}_{\tau,i}, i \in [n]} [f_1(\mathbf{x}_{\tau,i}; \boldsymbol{\theta}_{\tau-1}^1) + f_2(\mathbf{x}_{\tau,i}; \boldsymbol{\theta}_{\tau-1}^2)]$, given $(\mathbf{x}_{\tau,i}, r_{\tau,i}) \sim \mathcal{D}, i \in [n]$. Let $r_\tau$ be the corresponding reward. Let $(\mathbf{x}'_{\tau,i}, r'_{\tau,i}) \sim \mathcal{D}, i \in [n]$ be shadow samples from the same distribution and let $\mathbf{x}'_\tau = \arg\max_{\mathbf{x}'_{\tau,i}, i \in [n]} [f_1(\mathbf{x}'_{\tau,i}; \boldsymbol{\theta}_{\tau-1}^1) + f_2(\mathbf{x}'_{\tau,i}; \boldsymbol{\theta}_{\tau-1}^2)]$, with $r'_\tau$ being the corresponding reward. Then, we define

$$V_\tau := \mathop{\mathbb{E}}_{(\mathbf{x}'_{\tau,i}, r'_{\tau,i}), i \in [n]} \left[ \left| f_2(\mathbf{x}'_\tau; \widehat{\boldsymbol{\theta}}_{\tau-1}^2) - \left( r'_\tau - f_1(\mathbf{x}'_\tau; \widehat{\boldsymbol{\theta}}_{\tau-1}^1) \right) \right| \right]$$
$$- \left| f_2(\mathbf{x}_\tau; \widehat{\boldsymbol{\theta}}_{\tau-1}^2) - \left( r_\tau - f_1(\mathbf{x}_\tau; \widehat{\boldsymbol{\theta}}_{\tau-1}^1) \right) \right| . \tag{C.13}$$

Then, as $(\mathbf{x}_{\tau,i}, r_{\tau,i}) \sim \mathcal{D}, i \in [n]$, based on the definition of $(\mathbf{x}_\tau, r_\tau)$, we have

$$\mathbb{E}[V_\tau | \mathbf{F}_{\tau-1}] = \mathop{\mathbb{E}}_{(\mathbf{x}'_{\tau,i}, r'_{\tau,i}), i \in [n]} \left[ \left| f_2 \left( \mathbf{x}'_\tau; \widehat{\boldsymbol{\theta}}_{\tau-1}^2 \right) - \left( r'_\tau - f_1(\mathbf{x}'_\tau; \widehat{\boldsymbol{\theta}}_{\tau-1}^1) \right) \right| \mid \mathbf{F}_{\tau-1} \right]$$
$$- \mathop{\mathbb{E}}_{(\mathbf{x}_{\tau,i}, r_{\tau,i}), i \in [n]} \left[ \left| f_2 \left( \mathbf{x}_\tau; \widehat{\boldsymbol{\theta}}_{\tau-1}^2 \right) - \left( r_\tau - f_1(\mathbf{x}_\tau; \widehat{\boldsymbol{\theta}}_{\tau-1}^1) \right) \right| \mid \mathbf{F}_{\tau-1} \right] \tag{C.14}$$
$$= 0 ,$$

where $\mathbf{F}_{\tau-1}$ denotes the $\sigma$-algebra generated by the history $\mathcal{H}_{\tau-1} = \{\mathbf{x}_{\tau'}, r_{\tau'}\}_{\tau'=1}^{\tau-1}$. Moreover, we have

$$\frac{1}{t} \sum_{\tau=1}^t V_\tau = \frac{1}{t} \sum_{\tau=1}^t \mathop{\mathbb{E}}_{(\mathbf{x}'_{\tau,i}, r'_{\tau,i}), i \in [n]} \left[ \left| f_2(\mathbf{x}'_\tau; \widehat{\boldsymbol{\theta}}_{\tau-1}^2) - \left( r'_\tau - f_1(\mathbf{x}'_\tau; \widehat{\boldsymbol{\theta}}_{\tau-1}^1) \right) \right| \right]$$
$$- \frac{1}{t} \sum_{\tau=1}^t \left| f_2 \left( \mathbf{x}_\tau; \widehat{\boldsymbol{\theta}}_{\tau-1}^2 \right) - \left( r_\tau - f_1(\mathbf{x}_\tau; \widehat{\boldsymbol{\theta}}_{\tau-1}^1) \right) \right|$$

Since $\{V_\tau\}_{\tau=1}^t$ is a martingale difference sequence, inspired by Lemma 1 in (Cesa-Bianchi et al., 2001), applying the Hoeffding-Azuma inequality, with probability at least $1 - 3\delta$, we have

$$\mathbb{P} \left[ \underbrace{\frac{1}{t} \sum_{\tau=1}^t V_\tau - \frac{1}{t} \sum_{\tau=1}^t \mathbb{E}[V_\tau | \mathbf{F}_\tau]}_{I_1} > \underbrace{(1 + 2\xi)}_{I_2} \sqrt{\frac{2 \log(1/\delta)}{t}} \right] \leq \delta$$
$$\Rightarrow \qquad \mathbb{P} \left[ \frac{1}{t} \sum_{\tau=1}^t V_\tau > (1 + 2\xi) \sqrt{\frac{2 \log(1/\delta)}{t}} \right] \leq \delta , \tag{C.15}$$

where $I_1 = 0$ according to (C.14) and $I_2$ is because of (C.12).

According to Algorithm 1, $(\boldsymbol{\theta}_{t-1}^1, \boldsymbol{\theta}_{t-1}^2)$ is uniformly drawn from $\{(\widehat{\boldsymbol{\theta}}_\tau^1, \widehat{\boldsymbol{\theta}}_\tau^2)\}_{\tau=0}^{t-1}$. Thus, with probability $1 - 3\delta$, we have

$$
\begin{aligned}
&\mathbb{E}_{(\mathbf{x}_{t,i}', r_{t,i}'), i \in [n]} \mathbb{E}_{(\boldsymbol{\theta}_{t-1}^1, \boldsymbol{\theta}_{t-1}^2)} \left[ \left| f_2\left(\mathbf{x}_t'; \boldsymbol{\theta}_{t-1}^2\right) - \left(r_t' - f_1(\mathbf{x}_t'; \boldsymbol{\theta}_{t-1}^1)\right) \right| \right] \\
&= \frac{1}{t} \sum_{\tau=1}^t \mathbb{E}_{(\mathbf{x}_{\tau,i}', r_{\tau,i}'), i \in [n]} \left[ \left| f_2\left(\mathbf{x}_\tau'; \widehat{\boldsymbol{\theta}}_{\tau-1}^2\right) - \left(r_\tau' - f_1(\mathbf{x}_\tau'; \widehat{\boldsymbol{\theta}}_{\tau-1}^1)\right) \right| \right] \\
&\leq \underbrace{\frac{1}{t} \sum_{\tau=1}^t \left| f_2\left(\mathbf{x}_\tau; \widehat{\boldsymbol{\theta}}_{\tau-1}^2\right) - \left(r_\tau - f_1(\mathbf{x}_\tau; \widehat{\boldsymbol{\theta}}_{\tau-1}^1)\right) \right|}_{I_3} + (1 + 2\xi)\sqrt{\frac{2\log(1/\delta)}{t}} .
\end{aligned}
\tag{C.16}
$$

For $I_3$, according to Lemma C.6, for any $\widetilde{\boldsymbol{\theta}}^2$ satisfying $\|\widetilde{\boldsymbol{\theta}}^2 - \boldsymbol{\theta}_0^2\|_2 \leq \mathcal{O}(\frac{t^3}{\rho\sqrt{m}}\log m)$, with probability $1 - \delta$, we have

$$
\begin{aligned}
&\frac{1}{t} \sum_{\tau=1}^t \left| f_2(\mathbf{x}_\tau; \widehat{\boldsymbol{\theta}}_{\tau-1}^2) - \left(r_\tau - f_1(\mathbf{x}_\tau; \widehat{\boldsymbol{\theta}}_{\tau-1}^1)\right) \right| \\
&\leq \underbrace{\frac{1}{t} \sum_{\tau=1}^t \left| f_2(\mathbf{x}_\tau; \widetilde{\boldsymbol{\theta}}^2) - \left(r_\tau - f_1(\mathbf{x}_\tau; \widehat{\boldsymbol{\theta}}_{\tau-1}^1)\right) \right|}_{I_4} + \mathcal{O}\left(\frac{3L}{\sqrt{2t}}\right) .
\end{aligned}
\tag{C.17}
$$

For $I_4$, according to Lemma C.4 (1), these exists $\widetilde{\boldsymbol{\theta}}^2$ satisfying $\|\widetilde{\boldsymbol{\theta}}^2 - \boldsymbol{\theta}_0^2\|_2 \leq \mathcal{O}(\frac{t^3}{\rho\sqrt{m}}\log m)$, with probability $1 - \delta$, such that

$$
\begin{aligned}
&\frac{1}{t} \sum_{\tau=1}^t \left| f_2(\mathbf{x}_\tau; \widetilde{\boldsymbol{\theta}}^2) - \left(r_\tau - f_1(\mathbf{x}_\tau; \widehat{\boldsymbol{\theta}}_{\tau-1}^1)\right) \right| \\
&\leq \frac{1}{t}\sqrt{t} \sqrt{\underbrace{\sum_{\tau=1}^t \left( f_2(\mathbf{x}_\tau; \widetilde{\boldsymbol{\theta}}^2) - \left(r_\tau - f_1(\mathbf{x}_\tau; \widehat{\boldsymbol{\theta}}_{\tau-1}^1)\right) \right)^2}_{I_5}} \\
&\leq \frac{1}{\sqrt{t}} \sqrt{\underbrace{2\epsilon}_{I_5}} ,
\end{aligned}
\tag{C.18}
$$

where $I_5$ follows by a direct application of Lemma C.4 (1) by defining the loss $\mathcal{L}(\widetilde{\boldsymbol{\theta}}^2) = \frac{1}{2}\sum_{\tau=1}^t \left( f_2(\mathbf{x}_\tau; \widetilde{\boldsymbol{\theta}}^2) - \left(r_\tau - f_1(\mathbf{x}_\tau; \widehat{\boldsymbol{\theta}}_{\tau-1}^1)\right) \right)^2 \leq \epsilon$.

Combining Eq.(C.16), Eq.(C.17) and Eq.(C.18), with probability $(1 - 5\delta)$ we have

$$
\begin{aligned}
&\mathbb{E}_{(\mathbf{x}_{t,i}, r_{t,i}), i \in [n]} \left[ \left| f_2\left(\mathbf{x}_t; \boldsymbol{\theta}_{t-1}^2\right) - \left(r_t - f_1(\mathbf{x}_t; \boldsymbol{\theta}_{t-1}^1)\right) \right| \,\big|\, \{\mathbf{x}_\tau, r_\tau\}_{\tau=1}^{t-1} \right] \\
&\leq \sqrt{\frac{2\epsilon}{t}} + \mathcal{O}\left(\frac{3L}{\sqrt{2t}}\right) + (1 + 2\xi)\sqrt{\frac{2\log(1/\delta)}{t}} .
\end{aligned}
\tag{C.19}
$$

where the expectation over $(\boldsymbol{\theta}_{t-1}^1, \boldsymbol{\theta}_{t-1}^2)$ that is uniformly drawn from $\{(\widehat{\boldsymbol{\theta}}_\tau^1, \widehat{\boldsymbol{\theta}}_\tau^2)\}_{\tau=0}^{t-1}$.

Then, applying union bound to $t, n$ and rescaling the $\delta$ complete the proof. $\qquad\square$

**Corollary C.1.** *For any $\delta, \epsilon \in (0, 1), \rho \in (0, \mathcal{O}(\frac{1}{L}))$, suppose $m, \eta_1, \eta_2, K_1, K_2$ satisfy the conditions in Eq. (5.2) and $(\mathbf{x}_{\tau,i}, r_{\tau,i}) \sim \mathcal{D}, \forall \tau \in [t], i \in [n]$. For any $\tau \in [t]$, let*

$$
\mathbf{x}_\tau^* = \arg \max_{\mathbf{x}_{\tau,i}, i \in [n]} [h(\mathbf{x}_{\tau,i})],
$$

and $r_\tau^*$ is the corresponding reward, given $(\mathbf{x}_{\tau,i}, r_{\tau,i}), i \in [n]$. Then, with probability at least $(1 - \delta)$ over the random of the initialization, there exist $\boldsymbol{\theta}_{t-1}^{1,*}, \boldsymbol{\theta}_{t-1}^{2,*}$, s.t., $\|\boldsymbol{\theta}_{t-1}^{1,*} - \boldsymbol{\theta}_0^1\|_2 \leq \mathcal{O}(\frac{t^3}{\rho\sqrt{m}}\log m)$ and $\|\boldsymbol{\theta}_{t-1}^{2,*} - \boldsymbol{\theta}_0^2\|_2 \leq \mathcal{O}(\frac{t^3}{\rho\sqrt{m}}\log m)$, such that

$$\mathbb{E}_{(\mathbf{x}_{t,i}, r_{t,i}), i \in [n]} \left[\left|f_2\left(\phi(\nabla_{\boldsymbol{\theta}_{t-1}^{1,*}} f_1(\mathbf{x}_t^*; \boldsymbol{\theta}_{t-1}^{1,*})); \boldsymbol{\theta}_{t-1}^{2,*}\right) - \left(r_t^* - f_1(\mathbf{x}_t^*; \boldsymbol{\theta}_{t-1}^{1,*})\right)\right| \mid \{\mathbf{x}_\tau^*, r_\tau^*\}_{\tau=1}^{t-1}\right]$$

$$\leq \sqrt{\frac{2\epsilon}{t}} + \mathcal{O}\left(\frac{3L}{\sqrt{2t}}\right) + (1 + 2\xi)\sqrt{\frac{2\log(\mathcal{O}(tn/\delta))}{t}},$$

(C.20)

where the expectation is also taken over $(\boldsymbol{\theta}_{t-1}^{1,*}, \boldsymbol{\theta}_{t-1}^{2,*})$ that are uniformly drawn from $(\widehat{\boldsymbol{\theta}}_\tau^{1,*}, \widehat{\boldsymbol{\theta}}_\tau^{2,*}), \tau \in [t-1]$.

*Proof.* This a direct corollary of Lemma C.1, given the optimal historical pairs $\{\mathbf{x}_\tau^*, r_\tau^*\}_{\tau=1}^{t-1}$. For brevity, let $f_2(\mathbf{x}; \widehat{\boldsymbol{\theta}}_\tau^{2,*})$ represent $f_2\left(\phi(\nabla_{\widehat{\boldsymbol{\theta}}_\tau^{1,*}} f_1(\mathbf{x}; \widehat{\boldsymbol{\theta}}_\tau^{1,*})); \widehat{\boldsymbol{\theta}}_\tau^{2,*}\right)$.

Suppose that, for each $\tau \in [t-1]$, $\widehat{\boldsymbol{\theta}}_\tau^{1,*}$ and $\widehat{\boldsymbol{\theta}}_\tau^{2,*}$ are the parameters training on $\{\mathbf{x}_{\tau'}^*, r_{\tau'}^*\}_{\tau'=1}^\tau$ according to Algorithm 1. Note that these pairs $\{\mathbf{x}_{\tau'}^*, r_{\tau'}^*\}_{\tau'=1}^\tau$ are unknown to the algorithm we run, and the parameters $(\widehat{\boldsymbol{\theta}}_\tau^{1,*}, \widehat{\boldsymbol{\theta}}_\tau^{2,*})$ are not estimated. However, for the analysis, it is sufficient to show that there exist such parameters so that the conditional expectation of the error can be bounded.

In round $\tau \in [t]$, let $\mathbf{x}_\tau^* = \arg\max_{\mathbf{x}_{\tau,i} i \in [n]} [h(\mathbf{x}_{\tau,i})]$, given $(\mathbf{x}_{\tau,i}, r_{\tau,i}) \sim \mathcal{D}, i \in [n]$. Let $r_\tau^*$ be the corresponding reward. Let $(\mathbf{x}_{\tau,i}', r_{\tau,i}') \sim \mathcal{D}, i \in [n]$ be shadow samples from the same distribution and let $\mathbf{x}_\tau'^* = \arg\max_{\mathbf{x}_{\tau,i}', i \in [n]} h(\mathbf{x}_{\tau,i}')$, with $r_\tau'^*$ being the corresponding reward. Then, we define

$$V_\tau := \mathbb{E}_{(\mathbf{x}_{t,i}', r_{t,i}'), i \in [n]} \left[\left|f_2(\mathbf{x}_\tau'^*; \widehat{\boldsymbol{\theta}}_{\tau-1}^{2,*}) - \left(r_\tau'^* - f_1(\mathbf{x}_\tau'^*; \widehat{\boldsymbol{\theta}}_{\tau-1}^{1,*})\right)\right|\right]$$

$$- \left|f_2(\mathbf{x}_\tau^*; \widehat{\boldsymbol{\theta}}_{\tau-1}^{2,*}) - \left(r_\tau^* - f_1(\mathbf{x}_\tau^*; \widehat{\boldsymbol{\theta}}_{\tau-1}^{1,*})\right)\right|.$$

(C.21)

Then, as $(\mathbf{x}_{\tau,i}, r_{\tau,i}) \sim \mathcal{D}, i \in [n]$, we have

$$\mathbb{E}[V_\tau | \mathbf{F}_{\tau-1}] = \mathbb{E}_{(\mathbf{x}_{\tau,i}', r_{\tau,i}'), i \in [n]} \left[\left|f_2\left(\mathbf{x}_\tau'^*; \widehat{\boldsymbol{\theta}}_{\tau-1}^{2,*}\right) - \left(r_\tau'^* - f_1(\mathbf{x}_\tau'^*; \widehat{\boldsymbol{\theta}}_{\tau-1}^{1,*})\right)\right| \mid \mathbf{F}_{\tau-1}\right]$$

$$- \mathbb{E}_{(\mathbf{x}_{\tau,i}, r_{\tau,i}), i \in [n]} \left[\left|f_2\left(\mathbf{x}_\tau^*; \widehat{\boldsymbol{\theta}}_{\tau-1}^{2,*}\right) - \left(r_\tau^* - f_1(\mathbf{x}_\tau^*; \widehat{\boldsymbol{\theta}}_{\tau-1}^{1,*})\right)\right| \mid \mathbf{F}_{\tau-1}\right]$$

$$= 0,$$

(C.22)

where $\mathbf{F}_{\tau-1}$ denotes the $\sigma$-algebra generated by the history $\{\mathbf{x}_{\tau'}^*, r_{\tau'}^*\}_{\tau'=1}^{\tau-1}$.

Therefore, $\{V_\tau\}_{\tau=1}^t$ is a martingale difference sequence. Similarly, applying the Hoeffding-Azuma inequality to $V_\tau$, with probability $1 - 3\delta$, we have

$$\mathbb{E}_{(\mathbf{x}_{t,i}', r_{t,i}'), i \in [n]} \mathbb{E}_{(\boldsymbol{\theta}_{t-1}^{1,*}, \boldsymbol{\theta}_{t-1}^{2,*})} \left[\left|f_2\left(\mathbf{x}_t'^*; \boldsymbol{\theta}_{t-1}^{2,*}\right) - \left(r_t'^* - f_1(\mathbf{x}_t'^*; \boldsymbol{\theta}_{t-1}^{1,*})\right)\right|\right]$$

$$= \frac{1}{t}\sum_{\tau=1}^t \mathbb{E}_{(\mathbf{x}_{\tau,i}', r_{\tau,i}'), i \in [n]} \left[\left|f_2\left(\mathbf{x}_\tau'^*; \widehat{\boldsymbol{\theta}}_{\tau-1}^{2,*}\right) - \left(r_\tau'^* - f_1(\mathbf{x}_\tau'^*; \widehat{\boldsymbol{\theta}}_{\tau-1}^{1,*})\right)\right|\right]$$

$$\leq \underbrace{\frac{1}{t}\sum_{\tau=1}^t \left|f_2\left(\mathbf{x}_\tau^*; \widehat{\boldsymbol{\theta}}_{\tau-1}^{2,*}\right) - \left(r_\tau^* - f_1(\mathbf{x}_\tau^*; \widehat{\boldsymbol{\theta}}_{\tau-1}^{1,*})\right)\right|}_{I_3} + (1 + 2\xi)\sqrt{\frac{2\log(1/\delta)}{t}}.$$

(C.23)

For $I_3$, according to Lemma C.6, for any $\widetilde{\boldsymbol{\theta}}^{2,*}$ satisfying $\|\widetilde{\boldsymbol{\theta}}^{2,*} - \boldsymbol{\theta}_0^2\|_2 \leq \mathcal{O}(\frac{t^3}{\rho\sqrt{m}}\log m)$ , with probability $1 - \delta$, we have

$$
\begin{aligned}
&\frac{1}{t}\sum_{\tau=1}^{t}\left|f_2(\mathbf{x}_\tau^*; \widehat{\boldsymbol{\theta}}_{\tau-1}^{2,*}) - \left(r_\tau^* - f_1(\mathbf{x}_\tau^*; \widehat{\boldsymbol{\theta}}_{\tau-1}^{1,*})\right)\right| \\
&\leq \underbrace{\frac{1}{t}\sum_{\tau=1}^{t}\left|f_2(\mathbf{x}_\tau^*; \widetilde{\boldsymbol{\theta}}^{2,*}) - \left(r_\tau^* - f_1(\mathbf{x}_\tau^*; \widehat{\boldsymbol{\theta}}_{\tau-1}^{1,*})\right)\right|}_{I_4} + \mathcal{O}\left(\frac{3L}{\sqrt{2t}}\right) .
\end{aligned}
\tag{C.24}
$$

For $I_4$, according to Lemma C.4 (1), these exists $\widetilde{\boldsymbol{\theta}}^{2,*}$ satisfying $\|\widetilde{\boldsymbol{\theta}}^{2,*} - \boldsymbol{\theta}_0^2\|_2 \leq \mathcal{O}(\frac{t^3}{\rho\sqrt{m}}\log m)$, with probability $1 - \delta$, such that

$$
\begin{aligned}
&\frac{1}{t}\sum_{\tau=1}^{t}\left|f_2(\mathbf{x}_\tau^*; \widetilde{\boldsymbol{\theta}}^{2,*}) - \left(r_\tau^* - f_1(\mathbf{x}_\tau^*; \widehat{\boldsymbol{\theta}}_{\tau-1}^{1,*})\right)\right| \\
&\leq \frac{1}{t}\sqrt{t}\sqrt{\underbrace{\sum_{\tau=1}^{t}\left(f_2(\mathbf{x}_\tau^*; \widetilde{\boldsymbol{\theta}}^{2,*}) - \left(r_\tau^* - f_1(\mathbf{x}_\tau^*; \widehat{\boldsymbol{\theta}}_{\tau-1}^{1,*})\right)\right)^2}_{I_5}} \\
&\leq \frac{1}{\sqrt{t}}\sqrt{\underbrace{2\epsilon}_{I_5}} ,
\end{aligned}
\tag{C.25}
$$

where $I_5$ follows by a direct application of Lemma C.4 (1) by defining the loss $\mathcal{L}(\widetilde{\boldsymbol{\theta}}^{2,*}) = \frac{1}{2}\sum_{\tau=1}^{t}\left(f_2(\mathbf{x}_\tau^*; \widetilde{\boldsymbol{\theta}}^{2,*}) - \left(r_\tau^* - f_1(\mathbf{x}_\tau^*; \widehat{\boldsymbol{\theta}}_{\tau-1}^{1,*})\right)\right)^2 \leq \epsilon$. Combining above inequalities, with probability $(1 - 5\delta)$ we have

$$
\begin{aligned}
&\mathop{\mathbb{E}}_{(\mathbf{x}_{t,i}, r_{t,i}), i\in[n]}\left[\left|f_2\left(\mathbf{x}_t^*; \boldsymbol{\theta}_{t-1}^{2,*}\right) - \left(r_t^* - f_1(\mathbf{x}_t^*; \boldsymbol{\theta}_{t-1}^{1,*})\right)\right| \Big| \{\mathbf{x}_\tau^*, r_\tau^*\}_{\tau=1}^{t-1}\right] \\
&\leq \sqrt{\frac{2\epsilon}{t}} + \mathcal{O}\left(\frac{3L}{\sqrt{2t}}\right) + (1 + 2\xi)\sqrt{\frac{2\log(1/\delta)}{t}} .
\end{aligned}
\tag{C.26}
$$

Then, applying union bound to $t, n$ and rescaling the $\delta$ complete the proof.

$\square$

**Lemma C.2.** *Given* $\delta, \epsilon \in (0,1), \rho \in (0, \mathcal{O}(\frac{1}{L}))$, *suppose* $m, \eta_1, \eta_2, K_1, K_2$ *satisfy the conditions in Eq. (5.2). Then, with probability at least* $1 - \delta$, *in each round* $t \in [T]$, *for any* $\|\mathbf{x}\|_2 = 1$, *we have*

$$
\begin{aligned}
(1)\quad &|f_1(\mathbf{x}; \boldsymbol{\theta}_{t-1}^{1,*}) - f_1(\mathbf{x}; \boldsymbol{\theta}_{t-1}^1)| \\
&\leq \left(1 + \mathcal{O}\left(\frac{tL^3\log^{5/6}m}{\rho^{1/3}m^{1/6}}\right)\right)\mathcal{O}\left(\frac{Lt^3}{\rho\sqrt{m}}\log m\right) + \mathcal{O}\left(\frac{t^4L^2\log^{11/6}m}{\rho^{4/3}m^{1/6}}\right);
\end{aligned}
\tag{C.27}
$$

$$
\begin{aligned}
(2)\quad &\left|f_2\left(\phi(\nabla_{\boldsymbol{\theta}_{t-1}^{1,*}}f_1(\mathbf{x}; \boldsymbol{\theta}_{t-1}^{1,*})); \boldsymbol{\theta}_{t-1}^{2,*}\right) - f_2\left(\phi(\nabla_{\boldsymbol{\theta}_{t-1}^1}f_1(\mathbf{x}; \boldsymbol{\theta}_{t-1}^1)); \boldsymbol{\theta}_{t-1}^2\right)\right| \\
&\leq \left(1 + \mathcal{O}\left(\frac{tL^3\log^{5/6}m}{\rho^{1/3}m^{1/6}}\right)\right)\mathcal{O}\left(\frac{Lt^3}{\rho\sqrt{m}}\log m\right) + \mathcal{O}\left(\frac{t^4L^2\log^{11/6}m}{\rho^{4/3}m^{1/6}}\right);
\end{aligned}
\tag{C.28}
$$

$$
\begin{aligned}
(3)\quad &\|\nabla_{\boldsymbol{\theta}_{t-1}^1}f_1(\mathbf{x}; \boldsymbol{\theta}_{t-1}^1)\|_2, \|\nabla_{\boldsymbol{\theta}_{t-1}^2}f_2\left(\phi(\nabla_{\boldsymbol{\theta}_{t-1}^1}f_1(\mathbf{x}; \boldsymbol{\theta}_{t-1}^1)); \boldsymbol{\theta}_{t-1}^2\right)\|_2 \\
&\leq \left(1 + \mathcal{O}\left(\frac{tL^3\log^{5/6}m}{\rho^{1/3}m^{1/6}}\right)\right)\mathcal{O}(L) .
\end{aligned}
\tag{C.29}
$$

*Proof.* According to Lemma C.4 (2), $\|\widehat{\boldsymbol{\theta}}_{\tau-1}^1 - \boldsymbol{\theta}_0^1\|_2 \le \mathcal{O}(\frac{t^3}{\rho\sqrt{m}}\log m), \forall \tau \in [t]$. Thus, we have $\|\boldsymbol{\theta}_{t-1}^1 - \boldsymbol{\theta}_0^1\|_2 \le \mathcal{O}(\frac{t^3}{\rho\sqrt{m}}\log m)$.

First, based on Triangle inequality, for any $\|\mathbf{x}\|_2 = 1$, we have

$$
\begin{aligned}
\|\nabla_{\boldsymbol{\theta}_{t-1}^1} f_1(\mathbf{x}; \boldsymbol{\theta}_{t-1}^1)\|_2 &\le \|\nabla_{\boldsymbol{\theta}_0^1} f_1(\mathbf{x}; \boldsymbol{\theta}_0^1)\|_2 + \|\nabla_{\boldsymbol{\theta}_{t-1}^1} f_1(\mathbf{x}; \boldsymbol{\theta}_{t-1}^1) - \nabla_{\boldsymbol{\theta}_0^1} f_1(\mathbf{x}_i; \boldsymbol{\theta}_0^1)\|_2 \\
&\le \left(1 + \mathcal{O}\left(\frac{tL^3 \log^{5/6} m}{\rho^{1/3}m^{1/6}}\right)\right) \mathcal{O}(L)
\end{aligned}
\tag{C.30}
$$

where the last inequality is because of Lemma C.4 (3) and Lemma C.7.

Applying Lemma C.5 (1), for any $\mathbf{x} \sim \mathcal{D}, \|\mathbf{x}\|_2 = 1$ and $\|\boldsymbol{\theta}_{t-1}^{1,*} - \boldsymbol{\theta}_{t-1}^1\| \le \mathcal{O}(\frac{t^3}{\rho\sqrt{m}}\log m) = w$, we have

$$
\begin{aligned}
&|f_1(\mathbf{x}; \boldsymbol{\theta}_{t-1}^{1,*}) - f_1(\mathbf{x}; \boldsymbol{\theta}_{t-1}^1)| \\
\le& |\langle \nabla_{\boldsymbol{\theta}_{t-1}^1} f_1(\mathbf{x}_i; \boldsymbol{\theta}_{t-1}^1), \boldsymbol{\theta}_{t-1}^{1,*} - \boldsymbol{\theta}_{t-1}^1 \rangle| + \mathcal{O}(L^2\sqrt{m\log(m)})\|\boldsymbol{\theta}_{t-1}^{1,*} - \boldsymbol{\theta}_{t-1}^1\|_2 w^{1/3} \\
\le& \|\nabla_{\boldsymbol{\theta}_{t-1}^1} f_1(\mathbf{x}_i; \boldsymbol{\theta}_{t-1}^1)\|_2 \|\boldsymbol{\theta}_{t-1}^{1,*} - \boldsymbol{\theta}_{t-1}^1\|_2 + \mathcal{O}(L^2\sqrt{m\log(m)})\|\boldsymbol{\theta}_{t-1}^{1,*} - \boldsymbol{\theta}_{t-1}^1\|_2 w^{1/3} \\
\le& \left(1 + \mathcal{O}\left(\frac{tL^3 \log^{5/6} m}{\rho^{1/3}m^{1/6}}\right)\right) \mathcal{O}(\frac{Lt^3}{\rho\sqrt{m}}\log m) + \mathcal{O}\left(\frac{t^4 L^2 \log^{11/6} m}{\rho^{4/3}m^{1/6}}\right)
\end{aligned}
\tag{C.31}
$$

Similarly, we can use the same way to prove the lemmas for $f_2$. $\qquad\square$

**Lemma C.3.** *Let $f(\cdot; \widehat{\boldsymbol{\theta}}_t)$ follow the stochastic gradient descent of $f_1$ or $f_2$ in Algorithm 1. Suppose $m, \eta_1, \eta_2$ satisfy the conditions in Eq. (5.2). With probability at least $1 - \delta$, for any $\mathbf{x}$ with $\|\mathbf{x}\|_2 = 1$ and $t \in [T]$, it holds that*

$$
|f(\mathbf{x}; \widehat{\boldsymbol{\theta}}_t)| \le \mathcal{O}(1) + \mathcal{O}\left(\frac{t^3 nL \log m}{\rho\sqrt{m}}\right) + \mathcal{O}\left(\frac{t^4 nL^2 \log^{11/6} m}{\rho^{4/3}m^{1/6}}\right).
$$

*Proof.* Considering an inequality $|a - b| \le c$, we have $|a| \le |b| + c$. Let $\boldsymbol{\theta}_0$ be randomly initialized. Then applying Lemma C.5 (1), for any $\mathbf{x} \sim \mathcal{D}, \|\mathbf{x}\|_2 = 1$ and $\|\widehat{\boldsymbol{\theta}}_t - \boldsymbol{\theta}_0\| \le w$, we have

$$
\begin{aligned}
|f(\mathbf{x}; \widehat{\boldsymbol{\theta}}_t)| &\le |f(\mathbf{x}; \boldsymbol{\theta}_0)| + |\langle \nabla_{\boldsymbol{\theta}_0} f(\mathbf{x}_i; \boldsymbol{\theta}_0), \widehat{\boldsymbol{\theta}}_t - \boldsymbol{\theta}_0 \rangle| + \mathcal{O}(L^2\sqrt{m\log(m)})\|\widehat{\boldsymbol{\theta}}_t - \boldsymbol{\theta}_0\|_2 w^{1/3} \\
&\le \underbrace{\mathcal{O}(1)}_{I_0} + \underbrace{\|\nabla_{\boldsymbol{\theta}_0} f(\mathbf{x}_i; \boldsymbol{\theta}_0)\|_2 \|\widehat{\boldsymbol{\theta}}_t - \boldsymbol{\theta}_0\|_2}_{I_1} + \mathcal{O}(L^2\sqrt{m\log(m)})\|\widehat{\boldsymbol{\theta}}_t - \boldsymbol{\theta}_0\|_2 w^{1/3} \\
&\le \mathcal{O}(1) + \underbrace{\mathcal{O}(L) \cdot \mathcal{O}\left(\frac{t^3}{\rho\sqrt{m}}\log m\right)}_{I_2} + \underbrace{\mathcal{O}\left(L^2\sqrt{m\log(m)}\right) \cdot \mathcal{O}\left(\frac{t^3}{\rho\sqrt{m}}\log m\right)^{4/3}}_{I_3} \\
&= \mathcal{O}(1) + \mathcal{O}\left(\frac{t^3 L \log m}{\rho\sqrt{m}}\right) + \mathcal{O}\left(\frac{t^4 L^2 \log^{11/6} m}{\rho^{4/3}m^{1/6}}\right)
\end{aligned}
\tag{C.32}
$$

where: $I_0$ is based on the Lemma C.4 (3); $I_1$ is an application of Cauchy–Schwarz inequality; $I_2$ is according to Lemma C.4 (2) and (3) in which $\widehat{\boldsymbol{\theta}}_t$ can be considered as one step gradient descent; $I_3$ is due to Lemma C.4 (2).

Then, the proof is completed. $\qquad\square$

**Lemma C.4.** *Given a constant $0 < \epsilon < 1$, suppose $m$ satisfies the conditions in Eq. (5.2), the learning rate $\eta = \Omega(\frac{\rho}{poly(t,n,L)m})$, the number of iterations $K = \Omega(\frac{poly(t,n,L)}{\rho^2} \cdot \log \epsilon^{-1})$. Then, with probability at least $1 - \delta$, starting from random initialization $\boldsymbol{\theta}_0$,*

*(1) (Theorem 1 in (Allen-Zhu et al., 2019)) In round $t \in [T]$, given the collected data $\{\mathbf{x}_\tau, r_\tau\}_{i=\tau}^t$, the loss function is defined as: $\mathcal{L}(\boldsymbol{\theta}) = \frac{1}{2}\sum_{\tau=1}^t (f(\mathbf{x}_\tau; \boldsymbol{\theta}) - r_\tau)^2$. Then,*

*there exists $\widetilde{\boldsymbol{\theta}}$ satisfying $\|\widetilde{\boldsymbol{\theta}} - \boldsymbol{\theta}_0\|_2 \leq \mathcal{O}\left(\frac{t^3}{\rho\sqrt{m}}\log m\right)$, such that $\mathcal{L}(\widetilde{\boldsymbol{\theta}}) \leq \epsilon$ in $K = \Omega(\frac{poly(t,n,L)}{\rho^2} \cdot \log \epsilon^{-1})$ iterations;*

*(2) (Theorem 1 in (Allen-Zhu et al., 2019)) For any $k \in [K]$, it holds uniformly that $\|\boldsymbol{\theta}_t^{(k)} - \boldsymbol{\theta}_0\|_2 \leq \mathcal{O}\left(\frac{t^3}{\rho\sqrt{m}}\log m\right)$;*

*(3) Following the initialization, given $\|\mathbf{x}\|_2 = 1$, it holds that*

$$\|\nabla_{\boldsymbol{\theta}_0} f(\mathbf{x}; \boldsymbol{\theta}_0)\|_2 \leq \mathcal{O}(L), \quad |f(\mathbf{x}; \boldsymbol{\theta}_0)| \leq \mathcal{O}(1)$$

*where $\boldsymbol{\theta}_t^{(k)}$ represents the parameters of $f$ after $k \in [K]$ iterations of gradient descent in round $t$.*

*Proof.* Note that the output dimension $d$ in (Allen-Zhu et al., 2019) is removed because the output of network function in this paper always is a scalar. For (1) and (2), the only different setting from (Allen-Zhu et al., 2019) is that the initialization of last layer $\mathbf{W}_L \sim \mathcal{N}(0, \frac{2}{m})$ in this paper while $\mathbf{W}_L \sim \mathcal{N}(0, \frac{1}{d})$ in (Allen-Zhu et al., 2019). Because $d = 1$ and $m > d$ here, the upper bound in (Allen-Zhu et al., 2019) still holds for $\mathbf{W}_L$: with probability at least $1 - \exp(-\Omega(m/L))$, $\|\mathbf{W}_L\|_F \leq \sqrt{m/d}$. Therefore, (1) and (2) still hold for the initialization of this paper.

For (3), based on Lemma 7.1 in Allen-Zhu et al. (2019), we have $|f(\mathbf{x}; \boldsymbol{\theta}_0)| \leq \mathcal{O}(1)$. Denote by $D$ the ReLU function. For any $l \in [L]$,

$$\|\nabla_{W_l} f(\mathbf{x}; \boldsymbol{\theta}_0)\|_F \leq \|\mathbf{W}_L D \mathbf{W}_{L-1} \cdots D \mathbf{W}_{l+1}\|_F \cdot \|D \mathbf{W}_{l+1} \cdots \mathbf{x}\|_F \leq \mathcal{O}(\sqrt{L})$$

where the inequality is according to Lemma 7.2 in Allen-Zhu et al. (2019). Therefore, we have $\|\nabla_{\boldsymbol{\theta}_0} f(\mathbf{x}; \boldsymbol{\theta}_0)\|_2 \leq \mathcal{O}(L)$. $\qquad\square$

**Lemma C.5** (Lemma 4.1, (Cao and Gu, 2019)). *For any $\delta \in (0,1)$, if $w$ satisfies*

$$\mathcal{O}(m^{-3/2} L^{-3/2} [\log(tnL^2/\delta)]^{3/2}) \leq w \leq \mathcal{O}(L^{-6}[\log m]^{-3/2}),$$

*then, with probability at least $1 - \delta$ over randomness of $\boldsymbol{\theta}_0$, for any $t \in [T]$, $\|\mathbf{x}\|_2 = 1$, and $\boldsymbol{\theta}, \boldsymbol{\theta}'$ satisfying $\|\boldsymbol{\theta} - \boldsymbol{\theta}_0\|_2 \leq w$ and $\|\boldsymbol{\theta}' - \boldsymbol{\theta}_0\|_2 \leq w$, it holds uniformly that*

$$|f(\mathbf{x}_i; \boldsymbol{\theta}) - f(\mathbf{x}_i; \boldsymbol{\theta}') - \langle \nabla_{\boldsymbol{\theta}'} f(\mathbf{x}_i; \boldsymbol{\theta}'), \boldsymbol{\theta} - \boldsymbol{\theta}' \rangle| \leq \mathcal{O}(w^{1/3} L^2 \sqrt{m \log(m)}) \|\boldsymbol{\theta} - \boldsymbol{\theta}'\|_2. \quad \text{(C.33)}$$

**Lemma C.6.** *For any $\delta > 0$, suppose*

$$m > \tilde{\mathcal{O}}\left(poly(T, n, \rho^{-1}, L, \log(1/\delta) \cdot e^{\sqrt{\log 1/\delta}})\right).$$

*Then, with probability at least $1 - \delta$, setting $\eta_2 = \Theta(\frac{t^5}{\delta^2\sqrt{2m}})$ for algorithm 1, for any $\widetilde{\boldsymbol{\theta}}^2$ satisfying $\|\widetilde{\boldsymbol{\theta}}^2 - \boldsymbol{\theta}_0^2\|_2 \leq \mathcal{O}(\frac{t^3}{\rho\sqrt{m}}\log m)$, it holds that*

$$\sum_{\tau=1}^{t} \left| f_2\left(\phi(\nabla_{\widehat{\boldsymbol{\theta}}_{\tau-1}^1} f_1(\mathbf{x}_\tau; \widehat{\boldsymbol{\theta}}_{\tau-1}^1)); \widehat{\boldsymbol{\theta}}_{\tau-1}^2\right) - \left(r_\tau - f_1(\mathbf{x}_\tau; \widehat{\boldsymbol{\theta}}_{\tau-1}^1)\right) \right|$$

$$\leq \sum_{\tau=1}^{t} \left| f_2\left(\phi(\nabla_{\widehat{\boldsymbol{\theta}}_{\tau-1}^1} f_1(\mathbf{x}_\tau; \widehat{\boldsymbol{\theta}}_{\tau-1}^1)); \widetilde{\boldsymbol{\theta}}^2\right) - \left(r_\tau - f_1(\mathbf{x}_\tau; \widehat{\boldsymbol{\theta}}_{\tau-1}^1)\right) \right| + \mathcal{O}\left(\frac{3L\sqrt{t}}{\sqrt{2}}\right)$$

*Proof.* This is a direct application of Lemma 4.3 in (Cao and Gu, 2019) by setting $R = \frac{t^3}{\rho}\log m$, $\epsilon = \frac{LR}{\sqrt{2\nu t}}$, and $\nu = \nu' R^2$, where $\nu'$ is some small enough absolute constant. We set $L_\tau(\widehat{\boldsymbol{\theta}}_{\tau-1}^2) = \left| f_2(\nabla_{\widehat{\boldsymbol{\theta}}_{\tau-1}^1} f_1; \widehat{\boldsymbol{\theta}}_{\tau-1}^2) - \left(r_\tau - f_1(\mathbf{x}_\tau; \widehat{\boldsymbol{\theta}}_{\tau-1}^1)\right) \right|$. Based on Lemma C.4 (2), for any $\tau \in [t]$, we have

$$\|\widehat{\boldsymbol{\theta}}_\tau^2 - \widehat{\boldsymbol{\theta}}_{\tau-1}^2\|_2 \leq \|\widehat{\boldsymbol{\theta}}_\tau^2 - \boldsymbol{\theta}_0\|_2 + \|\boldsymbol{\theta}_0 - \widehat{\boldsymbol{\theta}}_{\tau-1}^2\|_2 \leq \mathcal{O}(\frac{t^3}{\rho\sqrt{m}}\log m).$$

Table 2: Selection Criterion Comparison ($\mathbf{x}_t$: selected arm in round $t$).

| Methods | Selection Criterion |
|---|---|
| Neural Epsilon-greedy | With probability $1 - \epsilon$, $\mathbf{x}_t = \arg\max_{\mathbf{x}_{t,i}, i \in [n]} f_1(\mathbf{x}_{t,i}; \boldsymbol{\theta}^1)$; Otherwise, select $\mathbf{x}_t$ randomly. |
| NeuralTS (Zhang et al., 2021) | For $\mathbf{x}_{t,i}, \forall i \in [n]$, draw $\hat{r}_{t,i}$ from $\mathcal{N}(f_1(\mathbf{x}_{t,i}; \boldsymbol{\theta}^1), \sigma_{t,i}{}^2)$. Then, select $\mathbf{x}_{t,\hat{i}}$, $\hat{i} = \arg\max_{i \in [n]} \hat{r}_{t,i}$. |
| NeuralUCB (Zhou et al., 2020) | $\mathbf{x}_t = \arg\max_{\mathbf{x}_{t,i}, i \in [n]} \left( f_1(\mathbf{x}_{t,i}; \boldsymbol{\theta}^1) + \text{UCB}_{t,i} \right)$. |
| EE-Net (Our approach) | $\forall i \in [n]$, compute $f_1(\mathbf{x}_{t,i}; \boldsymbol{\theta}^1)$, $f_2\left(\nabla_{\boldsymbol{\theta}^1} f_1(\mathbf{x}_{t,i}; \boldsymbol{\theta}^1); \boldsymbol{\theta}^2\right)$ (Exploration Net). Then $\mathbf{x}_t = \arg\max_{\mathbf{x}_{t,i}i \in [n]} f_3(f_1, f_2; \boldsymbol{\theta}^3)$. |

Then, according to Lemma 4.3 in (Cao and Gu, 2019), then, for any $\widetilde{\boldsymbol{\theta}}^2$ satisfying $\|\widetilde{\boldsymbol{\theta}}^2 - \boldsymbol{\theta}_0^2\|_2 \leq \mathcal{O}(\frac{t^3}{\rho\sqrt{m}}\log m)$, there exist a small enough absolute constant $\nu'$, such that

$$\sum_{\tau=1}^{t} L_\tau(\widehat{\boldsymbol{\theta}}_{\tau-1}^2) \leq \sum_{\tau=1}^{t} L_\tau(\widetilde{\boldsymbol{\theta}}^2) + 3t\epsilon. \tag{C.34}$$

Then, replacing $\epsilon$ completes the proof. $\qquad\square$

**Lemma C.7** (Theorem 5, Allen-Zhu et al. (2019)). *For any $\delta \in (0, 1)$, if $w$ satisfies that*

$$\mathcal{O}(m^{-3/2}L^{-3/2}\max\{\log^{-3/2} m, \log^{3/2}(Tn/\delta)\}) \leq w \leq \mathcal{O}(L^{-9/2}\log^{-3} m), \tag{C.35}$$

*then, with probability at least $1 - \delta$, for all $\|\boldsymbol{\theta} - \boldsymbol{\theta}_0\|_2 \leq w$, we have*

$$\|\nabla_{\boldsymbol{\theta}} f(\mathbf{x}; \boldsymbol{\theta}) - \nabla_{\boldsymbol{\theta}_0} f(\mathbf{x}; \boldsymbol{\theta}_0)\|_2 \leq \mathcal{O}(\sqrt{\log m}w^{1/3}L^3)\|\nabla_{\boldsymbol{\theta}_0} f(\mathbf{x}; \boldsymbol{\theta}_0)\|_2. \tag{C.36}$$

## D  MOTIVATION OF EXPLORATION NETWORK

In this section, we list one gradient-based UCB from existing works (Ban et al., 2021; Zhou et al., 2020), which motivates our design of exploration network $f_2$. Let $g(\mathbf{x}_t; \boldsymbol{\theta}_t) = \nabla_{\boldsymbol{\theta}_t} f(\mathbf{x}_t; \boldsymbol{\theta}_t)$.

**Lemma D.1.** *(Lemma 5.2 in (Ban et al., 2021)). Given a set of context vectors $\{\mathbf{x}_t\}_{t=1}^{T}$ and the corresponding rewards $\{r_t\}_{t=1}^{T}$, $\mathbb{E}(r_t) = h(\mathbf{x}_t)$ for any $\mathbf{x}_t \in \{\mathbf{x}_t\}_{t=1}^{T}$. Let $f(\mathbf{x}_t; \boldsymbol{\theta})$ be the $L$-layers fully-connected neural network where the width is $m$, the learning rate is $\eta$, the number of iterations of gradient descent is $K$. Then, there exist positive constants $C_1, C_2, S$, such that if*

$$m \geq poly(T, n, L, \log(1/\delta) \cdot d \cdot e^{\sqrt{\log 1/\delta}}), \quad \eta = \mathcal{O}(TmL + m\lambda)^{-1}, \quad K \geq \widetilde{\mathcal{O}}(TL/\lambda),$$

*then, with probability at least $1 - \delta$, for any $\mathbf{x}_t \in \{\mathbf{x}_t\}_{t=1}^{T}$, we have the following upper confidence bound:*

$$|h(\mathbf{x}_t) - f(\mathbf{x}_t; \boldsymbol{\theta}_t)| \leq \gamma_1 \|g(\mathbf{x}_t; \boldsymbol{\theta}_t)/\sqrt{m}\|_{\mathbf{A}_t^{-1}} + \gamma_2 + \gamma_1\gamma_3 + \gamma_4, \tag{D.1}$$

*where*

$$\gamma_1(m, L) = (\lambda + t\mathcal{O}(L)) \cdot ((1 - \eta m\lambda)^{J/2}\sqrt{t/\lambda}) + 1$$

$$\gamma_2(m, L, \delta) = \|g(\mathbf{x}_t; \boldsymbol{\theta}_0)/\sqrt{m}\|_{\mathbf{A}_t'^{-1}} \cdot \left(\sqrt{\log\left(\frac{\det(\mathbf{A}_t')}{\det(\lambda\mathbf{I})}\right) - 2\log\delta} + \lambda^{1/2}S\right)$$

$$\gamma_3(m, L) = C_2 m^{-1/6}\sqrt{\log m}t^{1/6}\lambda^{-7/6}L^{7/2}, \quad \gamma_4(m, L) = C_1 m^{-1/6}\sqrt{\log m}t^{2/3}\lambda^{-2/3}L^3$$

$$\mathbf{A}_t = \lambda\mathbf{I} + \sum_{i=1}^{t} g(\mathbf{x}_t; \boldsymbol{\theta}_t)g(\mathbf{x}_t; \boldsymbol{\theta}_t)^\intercal/m, \quad \mathbf{A}_t' = \lambda\mathbf{I} + \sum_{i=1}^{t} g(\mathbf{x}_t; \boldsymbol{\theta}_0)g(\mathbf{x}_t; \boldsymbol{\theta}_0)^\intercal/m.$$

Note that $g(\mathbf{x}_t; \boldsymbol{\theta}_0)$ is the gradient at initialization, which can be initialized as constants. Therefore, the above UCB can be represented as the following form for exploitation network $f_1$: $|h(\mathbf{x}_{t,i}) - f_1(\mathbf{x}_{t,i}; \boldsymbol{\theta}_t^1)| \leq \Psi(g(\mathbf{x}_t; \boldsymbol{\theta}_t))$.

Table 3: Exploration Direction Comparison.

| Methods | "Upward" Exploration | "Downward" Exploration |
|---------|---------------------|------------------------|
| NeuralUCB | $\checkmark$ | $\times$ |
| NeuralTS | Randomly | Randomly |
| EE-Net | $\checkmark$ | $\checkmark$ |

**EE-Net has smaller approximation error.** Given an arm $x$, let $f_1(x)$ be the estimated reward and $h(x)$ be the expected reward. The exploration network $f_2$ in EE-Net is to learn $h(x) - f_1(x)$, i.e., the residual between expected reward and estimated reward, which is the ultimate goal of making exploration. There are advantages of using a network $f_2$ to learn $h(x) - f_1(x)$ in EE-Net, compared to giving a statistical upper bound for it such as NeuralUCB, (Ban et al., 2021), and NeuralTS (in NeuralTS, the variance $\nu$ can be thought of as the upper bound). For EE-Net, the approximation error for $h(x) - f_1(x)$ is caused by the genenalization error of the neural network (Lemma B.1. in the manuscript). In contrast, for NeuralUCB, (Ban et al., 2021), and NeuralTS, the approximation error for $h(x) - f_1(x)$ includes three parts. The first part is caused by ridge regression. The second part of the approximation error is caused by the distance between ridge regression and Neural Tangent Kernel (NTK). The third part of the approximation error is caused by the distance between NTK and the network function. Because they use the upper bound to make selections, the errors inherently exist in their algorithms. By reducing the three parts of the approximation errors to only the neural network convergence error, EE-Net achieves tighter regret bound compared to them (improving by roughly $\sqrt{\log T}$).

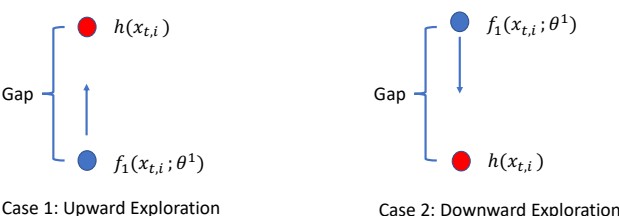

Case 1: Upward Exploration      Case 2: Downward Exploration

Figure 6: Two types of exploration: Upward exploration and Downward exploration. $f_1$ is the exploitation network (estimated reward) and $h$ is the expected reward.

**EE-Net has the ability to determine exploration direction.** The two types of exploration are described by Figure 6. When the estimated reward is larger than the expected reward, i.e., $h(x) - f_1(x) < 0$, we need to do the 'downward exploration', i.e., lowering the exploration score of $x$ to reduce its chance of being explored; when $h(x) - f_1(x) > 0$, we should do the 'upward exploration', i.e., raising the exploration score of $x$ to increase its chance of being explored. For EE-Net, $f_2$ is to learn $h(x) - f_1(x)$. When $h(x) - f_1(x) > 0$, $f_2(x)$ will also be positive to make the upward exploration. When $h(x) - f_1(x) < 0$, $f_2(x)$ will be negative to make the downward exploration. In contrast, NeuralUCB will always choose upward exploration, i.e., $f_1(x) + UCB(x)$ where $UCB(x)$ is always positive. In particular, when $h(x) - f_1(x) < 0$, NeuralUCB will further amplify the mistake. NeuralTS will randomly choose upward or downward exploration for all cases, because it draws a sampled reward from a normal distribution where the mean is $f_1(x)$ and the variance $\nu$ is the upper bound.

