# OpenReview forum: "EE-Net: Exploitation-Exploration Neural Networks in Contextual Bandits"
_ICLR.cc/2022/Conference — ICLR 2022 Spotlight_

### Official Review · Reviewer_BoiG · 2021-11-01

**Correctness:** 4
**Technical Novelty And Significance:** 3
**Empirical Novelty And Significance:** 4
**Recommendation:** 8
**Confidence:** 2

**Main Review:**

First of all, authors provided a very clear review on the various neural network based contextual bandits and illustrated their proposed approach very clearly through discussing the novelties they claim with their proposed technique in comparison with existing work in the area. The proposed technique has clear advantages over the existing literature and I believe it can be easily applied to existing domains that benefit from contextual bandit approached. The proposed method is well supported by theoretical findings and good empirical evidence. Authors also provide their source code and the datasets they use, which is a very important aspect for interested readers/researchers.

For the evaluation, I am not an expert on the contextual bandits but from educated guess, the experiment results are extensive, solid and impressive. They compared with existing state of art techniques using benchmark datasets and regret analysis together with ablation experiment results are presented.

I have a few comments/questions/concerns:
1.	Authors provide regret bounds’ complexity analysis for their technique. However, what are some concrete time complexity results for the proposed technique? My concern is, for online learning algorithms “time it takes to respond should be” quick, I am really curious to see how the techniques’ response time compares with the existing techniques response time in literature. Can authors provide this comparison analysis? Is there any overhead that the neural net based exploration brings to the solution?
2.	Parallel to 1st question, How does the solution scale with larger datasets?
3.	Figure 1 results are reported over 10 runs, I am curious why such a low number of runs are performed, not 30 for instance? Is it due to time-constraints or memory constraints for the machines used during experiments? Can you provide details for the machine co


**Summary Of The Paper:**

This paper presents a neural net based bandit approach with a novel exploration strategy. Specifically, the solution uses an exploitation network to estimate rewards for each arm and an exploration network to predict the potential gain compared to current reward estimate. It then uses a decision maker to select arms using one of the linear or nonlinear modes. The key differentiator/strength of the paper is using neural net based exploration strategy and demonstrating that the proposed solution has a tighter bound than the existing state-of-the-art bandit algorithms.

**Summary Of The Review:**

Overall, I liked the paper, very clearly written, organized. I am convinced with the novelty of the technique being proposed, I had a few concerns around some applicability issues in real life due to time complexity. I have detailed my questions in the previous box.

---

> ### Author Response · Authors · 2021-11-16
> **Response to Reviewer BoiG**
>
> Thanks very much for the detailed, constructive comments and questions.
>
> **Q1:**  What are some concrete time complexity results for the proposed technique?   Can authors provide this comparison analysis?
>
> **A1:** Thanks for raising this question. In the decision-making process, EE-Net only needs to compute the neural network outputs, which is very fast. In contrast, NeuralUCB and NeuralTS need to compute the gradient and update the gradient outer product matrix, whose time complexity is $O(p^2)$ ($p$ is the number of parameters in the neural network). Although in the training process (which can be done offline), NeuralUCB and NeuralTS need to train one neural network, while EE-Net needs to train two neural networks but the two neural networks can be trained in parallel.
>
> We've added additional experiments regarding the time cost (see details in **Appendix E** in the updated version).
> In summary, for decision-making time cost, NeuralUCB and NeuralTS are about 1.5 to 2 times slower than EE-Net, because NeuralUCB and NeuralTS need additional computational cost for the gradient matrix.
> For overall running time cost, EE-Net (without parallelization) is about 1.2-1.6 times slower than NeuralUCB and NeuralTS,
> because EE-Net needs to train two neural networks in each round while NeuralUCB and NeuralTS need to train one.
>
>
>
> **Q2:** How does the solution scale with larger datasets?
>
> **A2:**
> Thanks for your question.
> Because EE-Net can be considered as deep network models, it can easily scale to large data sets. We can use incremental gradient descent (SGD or Adam) to boost the training process in each round (this part can be moved offline), such that EE-Net can incrementally update with new incoming data.
>
>
> **Q3:**  Why run experiments 10 times?
>
> **A3:** Thanks for your suggestion.  In the experiments, we simply follow the setting of NeuralUCB and NeuralTS where they report the average of 10 runs.  We've added new experiments on Movielens and Yelp data sets where each method is run 30 times (see details in **Appendix E** in the updated version).
> These results are very similar to the results of running all the methods 10 times.

---

### Official Review · Reviewer_vjXZ · 2021-11-02

**Correctness:** 4
**Technical Novelty And Significance:** 4
**Empirical Novelty And Significance:** 3
**Recommendation:** 8
**Confidence:** 3

**Main Review:**

Strengths:

1. The studied neural contextual bandit is very interesting and important in the area of applying neural networks to online learning.
2. The proposed algorithm uses a novel and interesting network architecture, EE-Net, which is different from prior UCB-based [Zhou et al. 2020] or Thompson Sampling [Zhang et al. 2020] based neural bandit algorithms.
3. The authors give rigorous theoretical analysis and tighter regret bounds. While I do not think an improvement of \sqrt{log T} in the regret bound is significant, this does demonstrate the theoretical soundness of EE-Net.
4. The experiments are conducted on extensive real-world datasets, and the empirical results validate the superiority of the proposed EE-Net.


Weakness:

It would be better if the authors can highlight the technical challenges in regret analysis and compare their analytical techniques with prior neural contextual bandit works.


---After Rebuttal---

I have read the authors' response. They clearly explained the technical challenges of this paper, which well addressed my concerns. I appreciate that their algorithm only stores and updates the $O(p)$-dimensional parameter in neural networks, and does not need to explicitly maintain extra $O(p)$-dimensional vectors or $O(p \times p)$-dimensional matrix. This point is very good, and I have never seen it before in the neural contextual bandit literature. The theoretical improvement of removing the $d$ or $\tilde{d}$ is also interesting.
I am glad to increase my score from 6 to 8.

**Summary Of The Paper:**

This paper studies the neural contextual bandit problem, and proposes a neural-based bandit approach with a novel exploration strategy, called EE-Net. Besides utilizing a neural network (Exploitation network) to learn the reward function, EE-Net also uses another neural network (Exploration network) to adaptively learn potential gains compared to currently estimated reward. Then, a decision-maker is
constructed to combine the outputs from the Exploitation and Exploration networks. Theoretical guarantees and extensive experiments are provided to demonstrate the performance superiority of EE-Net over existing linear and neural bandit algorithms.


**Summary Of The Review:**

Overall, I think that the studied neural contextual bandit problem in this paper is well-motivated. The idea behind the proposed algorithm is novel and effective. Tighter theoretical guarantees and extensive experimental results are provided to demonstrate the superiority of the proposed algorithm. Therefore, I recommend “weak accept”.

---

> ### Author Response · Authors · 2021-11-16
> **Response to Reviewer vjXZ**
>
> Thanks very much for the positive comments and constructive suggestions.
> In the updated version, we've highlighted the technical challenges that EE-Net tackled or alleviated in the methodology perspective and regret analysis (see **Remark 4.2, 4.3, 5.1, 5.2** in the updated version).
> The key challenges can be summarized as follows:
>
> Methodology perspective:
> 1. **Challenge 1: Smaller approximation error to the potential gain $h(x) - f_1(x)$.**
>     To upper bound $h(x) - f_1(x)$,  NeuralUCB and NeuralTS need to calculate the (1) distance between $h(x)$ and ridge regression, (2) distance between ridge regression and NTK, and (3) distance between NTK and $f_1(x)$. In contrast, EE-Net uses a neural network to approximate $h(x) - f_1(x)$ and the error is only caused by the convergency of neural network. In the end, EE-Net lowers the approximation error by $\sqrt{\log T}$ roughly.
>
> 2. **Challenge 2: Ability of determining exploration direction.**  Given an arm $x$, when the estimated reward $f_1(x)$ is *lower* than  the expected reward $h(x)$, the learner should do the "upward" exploration, i.e., increase the chance of $x$ being explored; When $f_1(x)$ is *higher* than  $h(x)$, the learner should do the "downward" exploration, i.e., decrease the chance of $x$ being explored. EE-Net uses a neural network to learn $h(x) - f_1(x)$ (which has positive and negative scores) and has the ability to determine exploration direction. In contrast, NeuralUCB will always make "upward" exploration,  and NeuralTS will randomly choose between  "upward" exploration and "downward" exploration.
>
> 3. **Challenge 3: Space complexity.**  EE-Net reduces the space complexity from $O(p^2)$ to $O(p)$ ($p$ is the number of parameters of $f_1$).  NeuralUCB and NeuralTS have to maintain the gradient outer product matrix ($ \mathbb{R}^{p \times p}$) to store the historical gradients. On the contrary, EE-Net does not have this matrix and only regards gradient as the input of $f_2$.
>
>
> Regret analysis:
>
> 1. **Challenge 1 : Remove input dimension $d$.** In many real-world cases, the number of dimensions of the arm $d$ is a large number. To remove $d$, NeuralUCB and NeuralTS provide the effective dimension $\tilde{d}$. However, $\tilde{d}$ still can be very close to $d$ and affects the upper confidence bound and regret upper bound. In contrast, the regret bound of EE-Net does not have $d$ or $\tilde{d}$.
>
> 2. **Challenge 2 : Tighter regret upper bound.** EE-Net improves the regret upper bound over NeuralUCB/TS by $\sqrt{\log T}$ roughly.

---

> > ### Comment · Reviewer_vjXZ · 2021-11-21
> > **Comments after the Authors' Response**
> >
> > Thank you for your reply.
> >
> > My concerns on technical challenges and contributions are mostly addressed. The only thing is that the complexity $O(p)$ is still large since $p$ is large and the $\sqrt{\log T}$ improvement in regret is small.
> >
> > However, if my understanding is correct,  your algorithm does not need to explicitly maintain a $p$-dimensional vector or $p \times p$-dimensional matrix. Instead, your algorithm only stores the parameter vector in neural networks and executes the inference and training on neural networks. This point is good.

---

> > > ### Author Response · Authors · 2021-11-21
> > > **Response to Reviewer vjXZ (2)**
> > >
> > > Thanks for your response and we are glad that we addressed most of your concerns.
> > >
> > > (1) Yes, your understanding is accurate. Our approach only needs to store the parameters of the neural networks ($O(p)$) and execute the inference when making decisions. The training of neural networks can be conducted offline or in parallel.
> > >
> > > We believe that the space complexity $O(p)$ is appropriate for most real-world applications because other deep learning models also need $O(p)$ to store the parameters of the model, which have been shown to work well in many application domains such as image classification and machine translation.
> > >
> > > (2) The regret upper bound $O(\sqrt{T} \log T)$ has been around for decades in contextual bandits and the state-of-the-art approaches only can achieve this complexity. LinUCB[1,2], LinTS[3], NeuralUCB[4], and Neural[5] all achieve the regret bound $O(\sqrt{T} \log T)$. In this paper, our approach transforms the conventional ellipsoid-based exploration into the neural-based exploration strategy, achieving regret bound $O(\sqrt{T \log T})$ and removing $d$ or $\tilde{d}$.
> > > Although the improvement does not seem very large, we believe it is a significant step to push the boundary forward.
> > >
> > > Reference:
> > >
> > > [1] Varsha Dani, Thomas P. Hayes, and Sham M. Kakade. Stochastic linear optimization under bandit feedback. In Proceedings of the 21st Annual Conference on Learning Theory (COLT), 2008.
> > >
> > > [2] Abbasi-Yadkori, Yasin, Dávid Pál, and Csaba Szepesvári. "Improved algorithms for linear stochastic bandits." Advances in neural information processing systems 24 (2011).
> > >
> > > [3] Agrawal, Shipra, and Navin Goyal. "Thompson sampling for contextual bandits with linear payoffs." International Conference on Machine Learning. PMLR, 2013.
> > >
> > > [4] Zhou, Dongruo, Lihong Li, and Quanquan Gu. "Neural contextual bandits with ucb-based exploration." International Conference on Machine Learning. PMLR, 2020.
> > >
> > > [5] Zhang, W., Zhou, D., Li, L. and Gu, Q. "Neural Thompson Sampling." International Conference on Learning Representations. 2021.

---

### Official Review · Reviewer_KcQy · 2021-11-03

**Correctness:** 3
**Technical Novelty And Significance:** 3
**Empirical Novelty And Significance:** 2
**Recommendation:** 6
**Confidence:** 3

**Main Review:**

**Strengths**

The idea of utilizing neural networks to model potential reward gains for exploration is very interesting; it seems quite different from rule-based UCB and Thompson sampling.

**Weaknesses**

On the theoretical side, the assumptions on over-parameterized neural networks are utilized to prove the regret bound, which is tighter than NeuralUCB and NeuralTS. However, since the regret bounds of NeuralUCB and NeuralTS are derived from the assumptions of the neural tangent kernel, it is not clear whether the theoretical advantage comes from the method itself, or from the assumptions.

On the experimental side, it is not sufficiently rigorous to tune the hyper-parameters. It is said that

> For all grid-searched parameters, we choose the best of them for the comparison and report the averaged results of 10 runs for all methods.
>

In real applications, it is not possible to tune hyper-parameters in this way primarily due to the bandit feedback. It is necessary to avoid the risk of overfitting in the experimental setting.

**Summary Of The Paper:**

This paper studies neural network-based contextual bandits. Different from existing works which utilize UCB or Thompson sampling for exploration, the exploration component of the proposed method EE-Net is also modeled with a neural network.

Based on theoretical tools on over-parameterized neural networks, it is proved that EE-Net achieves the regret bound of $\mathcal{O}(\sqrt{T\log T})$. In contrast, existing neural network-based contextual bandit methods, such as NeuralUCB and NeuralTS, achieves the regret bound of $\mathcal{O}(\sqrt{T} \log T)$ based on the theoretical tool of the neural tangent kernel.

Experiments are conducted on four public datasets to validate effectiveness of EE-Net, compared with a variety of contextual bandit methods including both neural-based methods and linear contextual bandit methods.

**Summary Of The Review:**

The idea is interesting and as far as I know it is novel. However, both the theoretical and experimental results should be verified more carefully. See Main Review for details.

---

> ### Author Response · Authors · 2021-11-16
> **Response to Reviewer KcQy**
>
> Thanks very much for your constructive comments and questions.  Our responses are as follows.
>
> **Q1**: Does the theoretical advantage come from the method itself or the assumptions?
>
> **A1**: Thanks for this question.  The theoretical advantages come from the method itself. Given an arm $x$, let $f_1(x)$ be the estimated reward and $h(x)$ be the expected reward.  To estimate the potential gain $ h(x) - f_1(x)$, NeuralUCB and NeuralTS derive its upper bound (for NeuralTS, the variance $\nu$ can be thought of as the upper bound). The upper bound contains three parts of the approximation error. The first part is caused by ridge regression ( Eq.B.5 in NeuralUCB and Lemma 4.3 in NeuralTS). The second part of the approximation error is caused by the distance between ridge regression and Neural Tangent Kernel (NTK) (Lemma B.2 in NeuralUCB and Lemma C.1 in NeuralTS ). The third part of the approximation error is caused by the distance between NTK and the network function (Lemma B.4 in NeuralUCB and Lemma C.2 in NeuralTS). Because NeuralUCB and NeuralTS use the upper bound to make selections, the errors inherently exist in their algorithms.
>
> In contrast,  EE-Net uses a neural network to learn $h(x) - f_1(x)$. The approximation error is only caused by the convergence error of the neural network (Lemma B.1 in the manuscript). By reducing the three parts of the approximation error (NeuralUCB/TS) to only the neural network convergence error, EE-Net achieves tighter regret bound compared to them (improving by roughly $\sqrt{\log T}$).
>
> **Comment 2**: In real applications, it is not possible to tune hyper-parameters in this way primarily due to the bandit feedback.
>
> **A2**: Yes, this kind of evaluation can be tricky in real life, but we are just following what related recent work has done.  In the experimental evaluation, we try to find the best performance of the baselines. Therefore, we follow the experiment setting of NeuralUCB and NeuralTS where they did the grid-search and reported the average of 10 runs. For EE-Net, it doesn't have hyperparameters except for the network structures and the learning rate. Here, we use the same network structures and learning rate for EE-Net and the baselines to ensure a fair comparison. Therefore, in the experiments, we aim to provide useful insights for the application of EE-Net in real problems.

---

### Official Review · Reviewer_SiY7 · 2021-11-06

**Correctness:** 3
**Technical Novelty And Significance:** 2
**Empirical Novelty And Significance:** 2
**Recommendation:** 6
**Confidence:** 3

**Main Review:**

The paper is well written and outputs a better regret bound with an interesting idea. The reviewer has the one concern about the exploration network:
     the proposed exploration net is mainly based on the error bound derived from Ban et al. 2021. This paper uses neural networks to learn the complicated function. This step makes the reviewer confused. We already have an analytical equation (even it has some unknown constants) for that error bound (lemma D.1). What’s the point of training a neural network to learn it instead of learning the unknown parts in that equation? Also,  why do we use the gradients as the input for the neural network instead of putting more known items in the input, e.g., A_t in the lemma D.1 (does this give more info to the training process)?


**Summary Of The Paper:**

This paper proposes EE-NeT for contextual bandits which contains two neutral networks: one for learning the underlying reward function and another for learning the potential gains of arms if explored. They prove that  EE-NeT achieves better regret bounds than state-of-the-art neural bandits algorithms for both UCB-based and TS-based. This paper also shows that EE-NeT outperforms existing linear and neural bandit approaches.

**Summary Of The Review:**

The reviewer doubts the motivation of using neural networks to the confidence found when we have an analytical equation for that.

---

> ### Author Response · Authors · 2021-11-15
> **Response to Reviewer SiY7 (1)**
>
> Thanks very much for your constructive comments and questions. Our responses to raised questions and concerns are as follows.
>
> **Q1**: What's the point of using a neural network to learn the unknown parts of the inequation?
>
> **A1**: Thanks for raising this question. Given an arm $x$, let $f_1(x)$ be the estimated reward and $h(x)$ be the expected reward. The exploration network $f_2$ in EE-Net is to learn $h(x) - f_1(x)$, i.e., the residual between expected reward and estimated reward, which is the ultimate goal of making exploration. There are three advantages of using a network $f_2$ to learn $h(x) - f_1(x)$ in EE-Net, compared to giving a statistical upper bound for it such as NeuralUCB, [Ban et al. 2021], and NeuralTS (in NeuralTS, the variance $\nu$ can be thought of as the upper bound). Because [Ban et al. 2021] focuses on complex networks (multiple bandits), we didn't include it in the comparison.
>
>
> 1. **EE-Net has smaller error to approximate $ h(x) - f_1(x)$**. For EE-Net, the approximation error for $h(x) - f_1(x)$ is caused by the convergence error of the neural network (Lemma B.1. in the manuscript). In contrast, for NeuralUCB, [Ban et al. 2021], and NeuralTS, the approximation error for $h(x) - f_1(x)$ includes three parts.
>     The first part is caused by ridge regression (Eq.B.5 in NeuralUCB, Page 10 in [Ban et al. 2021], and Lemma 4.3 in NeuralTS). The second part of the approximation error is caused by the distance between ridge regression and Neural Tangent Kernel (NTK) (Lemma B.2 in NeuralUCB, Lemma 8.7 in [Ban et al. 2021], and Lemma C.1 in NeuralTS). The third part of the approximation error is caused by the distance between NTK and the network function (Lemma B.4 in NeuralUCB, Lemma 8.5 in [Ban et al. 2021],  and Lemma C.2 in NeuralTS). Because they use the upper bound to make selections, the errors inherently exist in their algorithms.
>     By reducing the three parts of the approximation errors to only the neural network convergence error, EE-Net achieves tighter regret bound compared to them (improving by roughly $\sqrt{\log T}$).
>
> 2. **EE-Net has the ability to determine exploration direction.** When the estimated reward is larger than the expected reward, i.e., $h(x) - f_1(x) < 0$, we need to do the "downward" exploration, i.e., lowering the exploration score of $x$ to reduce its chance of being explored; When $h(x) - f_1(x) > 0$, we should do the "upward exploration", i.e., raising the exploration score of $x$ to increase its chance of being explored. For EE-Net, $f_2$ is to learn $h(x) - f_1(x)$. When $h(x) - f_1(x) > 0$, $f_2(x)$ will also be positive to make the upward exploration. When  $h(x) - f_1(x) < 0$, $f_2(x)$ will be negative to make the downward exploration. In contrast, NeuralUCB will always choose upward exploration, i.e., $f_1(x)+UCB(x)$ where $UCB(x)$ is always positive. In particular, when $h(x) - f_1(x) < 0$, NeuralUCB will further amplify the mistake. NeuralTS will randomly choose upward or downward exploration for all cases, because it draws a sampled reward from a normal distribution where the mean is $f_1(x)$ and the variance $\nu$ is the upper bound.
>
> 3. **EE-Net reduces the space complexity from $O(p^2)$ to $O(p)$ ($p$ is the number of parameters of $f_1$).** Let $g(x, \theta)$ be the gradient $\nabla_{\theta_1}f_1(x; \theta_1)$.
>     NeuralUCB and NeuralTS have to maintain the gradient outer product matrix $g(x, \theta)g(x, \theta)^\top \in \mathbb{R}^{p \times p}$ to store the historical gradients ($\mathbf{Z}$ in NeuralUCB,  $\mathbf{U}$ in NeuralTS).
>     On the contrary, EE-Net does not have this matrix and only regards $g(x, \theta)$ as the input of $f_2$. Note that $p$ usually is a large number ($p > 10^6$ for VGG in some image classification tasks).
>
>
> Therefore, there are three advantages of using a neural network to learn the potential gain $h(x) - f_1(x)$, which motivated us to design EE-Net in this way.

---

> > ### Author Response · Authors · 2021-11-16
> > **Response to Reviewer SiY7 (2)**
> >
> > **Q2**: Why do we use the gradient as the input of $f_2$?
> >
> > **A2**: Thanks for raising this question.
> > We use the gradient $\nabla_{\theta_1}f_1(x; \theta_1)$ as input because $\nabla_{\theta_1}f_1$ incorporates two aspects of information: the arm features and the discriminative information of $f_1$ with respect to the arm. When we make exploration, we need to consider the discriminative ability/accuracy of exploitation neural network $f_1$.
> > Therefore, we regard $\nabla_{\theta_1}f_1(x; \theta_1)$ as input.
> >
> > Moreover, in the upper bound of NeuralUCB or the variance of NeuralTS, there is a recursive term $A_{t-1} = I + \sum_{i=1}^{t-1} \nabla_{\theta_1}f_1(x; \theta_1^i) \nabla_{\theta_1}f_1(x; \theta_1^i)^\top$ which is a function of past gradients up to $(t-1)$ and incorporates relevant historical information, and the new variable is the current gradient $\nabla_{\theta_1}f_1(x; \theta_1^t)$. On the contrary, in EE-Net, the recursive term which depends on past gradients is $\theta^2_{t-1}$ in the exploration network $f_2$ because we have conducted gradient descent on $\theta^2_{t-1}$,  and the new input is current gradient $\nabla_{\theta_1}f_1(x; \theta_1^t)$. Therefore, this form is similar to neuralUCB/TS, but EE-net does not (need to) make a specific assumption about the functional form of past gradients, and is also more memory-efficient.
> >
> > **Why don't use $\mathbf{A}_t$ as input?**   First,  $A_t$  is the gradient outer product matrix $A_t \in \mathbb{R}^{p \times p}$, which can easily cause dimension explosion. Second, consider the historical data  \{$  A_t  | t \in [T] $\} .
> > For any  $t \in [T] $, $A_t - A_{t-1} = \nabla_{\theta_1^t}f_1(x; \theta_1^t) \nabla_{\theta_1^t}f_1(x; \theta_1^t)^\top$. This indicates that the incremental information of $A_t$ of each round corresponds to $\nabla_{\theta_1}f_1(x; \theta_1^t)$. Therefore, we can directly use the gradient  {$\nabla_{\theta_1}f_1(x; \theta_1^i) | i \in [t] $} as the training set, avoiding the dimension explosion.

---

> > ### Comment · Reviewer_SiY7 · 2021-11-22
> > **Comments after the Authors' Response**
> >
> > Thanks for your detailed reply which perfectly answers my questions about the motivation of EE-Net, also the choice of input for exploration net.

---

> > > ### Author Response · Authors · 2021-11-22
> > > **Response to Reviewer SiY7 (3)**
> > >
> > > Thanks so much for your time and effort spent on the review, and we are very glad that we addressed your concerns.

---

### Decision · Program_Chairs · 2022-01-20

**Decision:**

Accept (Spotlight)

**Comment:**

Summary: This paper studies the neural contextual bandit problem, and proposes a neural-based bandit approach with a novel exploration strategy, called EE-Net. Besides utilizing a neural network (Exploitation network) to learn the reward function, EE-Net also uses another neural network (Exploration network) to adaptively learn potential gains compared to currently estimated reward.

Discussions: The reviewers appreciated the novelty and the quality of the ideas and results in this paper. Most questions were about details in algorithm design choices and in the analysis. The authors have addressed these questions and updated their draft. The reviewers have now reached a consensus and recommend accepting this paper.

Recommendation: Accept.